# Distinct neuronal types contribute to hybrid temporal encoding strategies in primate auditory cortex

Xiao-Ping Liu ◉*, Xiaoqin Wang ◉*

Laboratory of Auditory Neurophysiology, Department of Biomedical Engineering, Johns Hopkins University School of Medicine, Baltimore, Maryland, United States of America

* xpliu16@gmail.com (X-PL); xiaoqin.wang@jhu.edu (XW)

## Abstract

Studies of the encoding of sensory stimuli by the brain often consider recorded neurons as a pool of identical units. Here, we report divergence in stimulus-encoding properties between subpopulations of cortical neurons that are classified based on spike timing and waveform features. Neurons in auditory cortex of the awake marmoset (*Callithrix jacchus*) encode temporal information with either stimulus-synchronized or nonsynchronized responses. When we classified single-unit recordings using either a criteria-based or an unsupervised classification method into regular-spiking, fast-spiking, and bursting units, a subset of intrinsically bursting neurons formed the most highly synchronized group, with strong phase-locking to sinusoidal amplitude modulation (SAM) that extended well above 20 Hz. In contrast with other unit types, these bursting neurons fired primarily on the rising phase of SAM or the onset of unmodulated stimuli, and preferred rapid stimulus onset rates. Such differentiating behavior has been previously reported in bursting neuron models and may reflect specializations for detection of acoustic edges. These units responded to natural stimuli (vocalizations) with brief and precise spiking at particular time points that could be decoded with high temporal stringency. Regular-spiking units better reflected the shape of slow modulations and responded more selectively to vocalizations with overall firing rate increases. Population decoding using time-binned neural activity found that decoding behavior differed substantially between regular-spiking and bursting units. A relatively small pool of bursting units was sufficient to identify the stimulus with high accuracy in a manner that relied on the temporal pattern of responses. These unit type differences may contribute to parallel and complementary neural codes.

## Introduction

Neuronal type is often not considered in auditory cortical electrophysiology studies, particularly in primates where cell type–specific markers and tools are not widely available. The heterogeneity of extracellular action potential morphology, which depends in part on the location of the electrode relative to the unit [1–3], presents an additional challenge. Nevertheless, recent

**Data Availability Statement:** All relevant data are within the paper and its Supporting Information files.

**Funding:** This work was supported by National Institutes of Health grants DC003180 and

DC005808 to XQW. The funders had no role in study design, data collection and analysis, decision to publish, or preparation of the manuscript.

**Competing interests:** The authors have declared that no competing interests exist.

**Abbreviations:** AIC, Akaike information criterion; BIC, Bayesian information criterion; Bu, bursting; CI, correlation index; CV, coefficient of variation; FRB, fast rhythmic bursting; FS, fast-spiking; GMM, Gaussian mixture model; IB, intrinsic bursting; ISI, interspike interval; LDA, linear discriminant analysis; MCC, maximum correlation coefficient; NCM, caudomedial nidopallium; PCA, principal component analysis; PSTH, peristimulus time histogram; RS, regular-spiking; SAM, sinusoidal amplitude modulation; STRF, spectro-temporal response function; VS, vector strength.

studies have classified cortical extracellular units in detail and demonstrated the functional relevance of these types [4–7]. Combined transcriptomic, morphological, and electrophysiological approaches are also revealing the significant diversification of even excitatory neurons in cortex [8,9].

Neurons in primate auditory cortex have diverse stimulus response properties, but it is unclear what aspects of this diversity are accounted for by neuronal type. For instance, they may encode click trains [10] or amplitude modulation [11] with either stimulus-synchronized or nonsynchronized responses (reviewed in [12]). While visual objects can be recognized from static images, temporal features are critical to auditory object recognition. Speech intelligibility is supported by temporal envelope cues [13] and temporal features such as voice onset time [14,15], while beats, rhythmic patterns, and expressive timing are integral to music [16]. Timing of discrete landmarks, such as acoustic edges, may be a primary form of representation of speech in humans [17,18].

To examine the contribution of neuronal type to temporal coding properties, we partitioned extracellular single-unit recordings from marmoset auditory cortex using 2 methods, one relying on criteria chosen by inspection and one based on unsupervised clustering on a set of informative features. Both methods labeled most but not all units, with strong agreement between them. Regular-spiking (RS) units formed the largest group, with substantial minorities being fast spiking (FS) or bursting (Bu). Bursting has long been noticed as a property of some excitatory cortical neurons [6,7,9,19–27], but their role in the auditory cortex has not been examined. We find that the temporal dynamics of responses to synthetic and natural stimuli are strongly influenced by unit type, with a subset of bursting units corresponding to the most synchronized type. These units acted as differentiators that phase-locked well to acoustic edges and fast modulations, a behavior described in biophysical models of bursting neurons [28]. Similarly to neurons in higher auditory areas in songbirds [29,30], they encoded vocalizations with transient and precise firing at particular times during the calls. Furthermore, their spike trains could be used to decode call identity with high spike timing stringency. In contrast, the more selective and sustained responses of RS units might contribute better to a labeled-line rate code. Correspondingly, we find that a relatively small pool of bursting units was sufficient to achieve high population decoding performance if fine temporal information is preserved. This result is reminiscent of a previous report in macaque auditory cortex of a group of highly temporally precise units with privileged contributions to sensory decoding [31]. Our results support the notion that auditory cortex uses varied transformations in diverse neuronal types to encode dynamic stimuli with both rate and temporal codes.

## Results

### Classification of unit types

Single-unit tungsten microelectrode recordings were obtained in core auditory cortex of 2 marmosets (Materials and methods and S2A, S2D, and S2E Fig). The rate and sound of spikes on the audio monitor appeared to distinguish RS-like and FS-like units. Sometimes, a third pattern was observed of intermittent bursts or doublets. An example of each unit type is shown in Fig 1. The FS unit had a higher spontaneous and driven rate (Fig 1A and 1E) and narrower spike waveform (Fig 1D, left of each pair). The bursting unit average spike waveform contained a second spikelet (reduced in amplitude in part due to temporal smearing). Typically, neurons in awake marmoset auditory cortex produce a more sustained response to more optimized stimuli [32]. Synthetic bandpass noise or tones at the frequency, sound level, and bandwidth that produced the maximal driven firing rate (referred to as best frequency, level, and bandwidth) evoked robust sustained responses in the RS and FS units, while less optimal

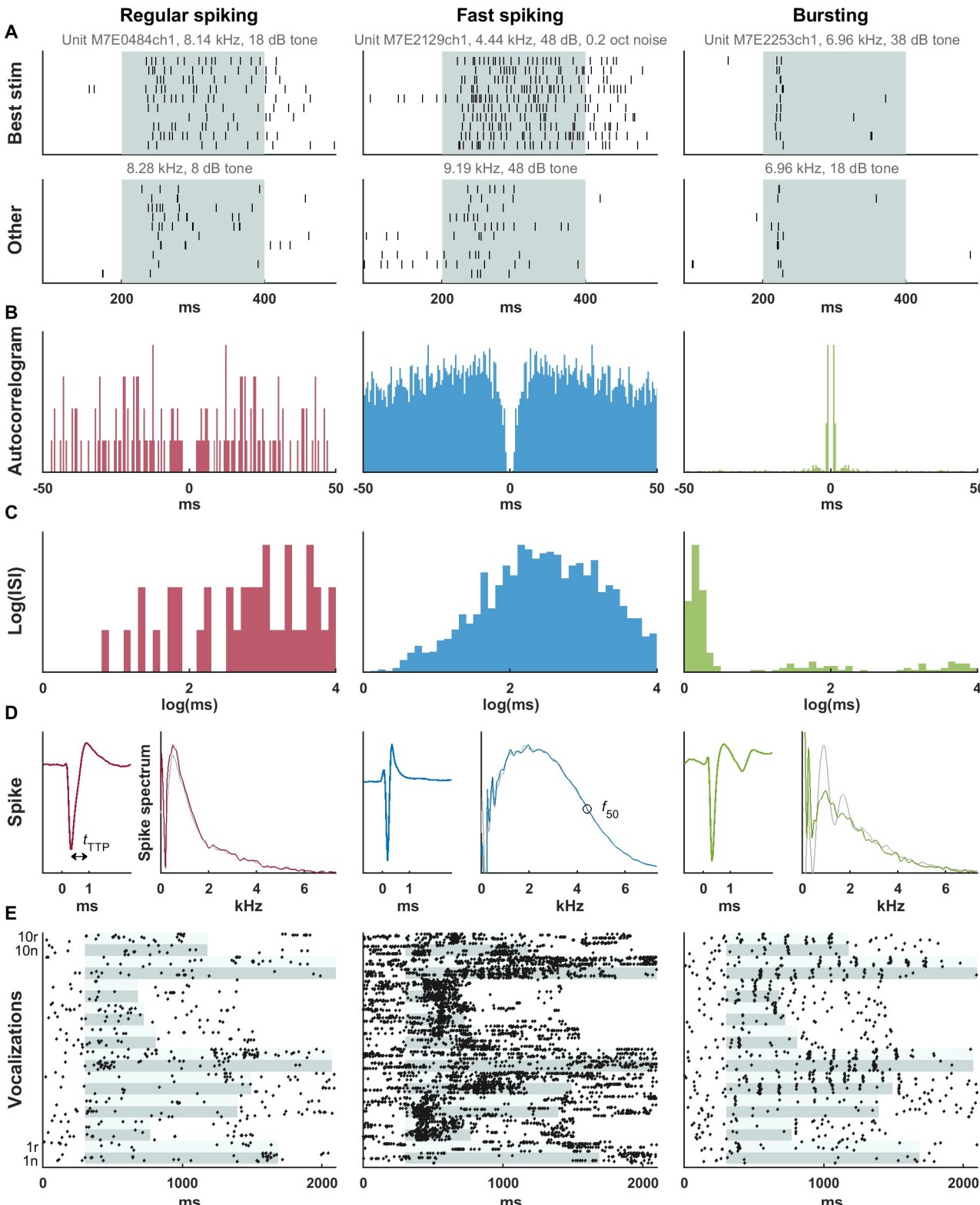

**Fig 1. Examples of 3 unit types discerned from extracellular recordings—RS, FS, and Bu.** The RS and FS units had relatively sustained responses at best frequency, intensity, and bandwidth, and less sustained responses to less optimal stimuli. However, the Bu unit responded transiently even to the relatively optimized simple stimulus (A) as well as conspecific vocalizations (E). Stimulus duration is shown in light aqua, with each band encompassing 10 trials; adjacent stimuli in (E) are shown in alternating shades. The Bu unit had a peak at a few milliseconds on the autocorrelogram

(B) and on the histogram of the log of ISIs (C) corresponding to an intraburst frequency of 769 Hz. The FS unit had higher spontaneous and driven rates, a narrower spike waveform (D, left of pair), and more high-frequency content in the spike waveform (D, right of pair). For analysis, only spikes not closely preceded or followed by another spike were included in the average. However, in (D), we allowed closely following spikes to illustrate the bursting tendency; the associated spectrum (light gray line) shows periodic peaks corresponding to harmonics of the intraburst frequency. These fluctuations are mostly, but not completely, suppressed by disallowing closely spaced spikes (green line). Two measures of waveform width are indicated in (D)—the trough-to-peak time ($t_{TTP}$), and the frequency at which the spike spectrum falls to 50% of its peak value ($f_{50}$). (E) Responses of the same 3 units to natural ("n") and time-reversed ("r") vocalization tokens (see S6A Fig for spectrograms). Data underlying this figure can be found in S1 Data. Bu, bursting; FS, fast-spiking; ISI, interspike interval; RS, regular-spiking.

stimuli produced less sustained responses (Fig 1A). However, the bursting unit had a short-latency, transient response even after optimizing those stimulus parameters. When presented with prerecorded conspecific vocalizations, the bursting unit responded transiently at particular moments throughout (Fig 1E).

To formally classify unit types, we first used a criteria-based method that resulted in 297 RS, 92 FS, 97 bursting, and 105 unclassified units. To separate bursting from nonbursting neurons, we used spike-timing analysis of frequency tuning protocols, including the entire stimulus and non-stimulus period, but later confirm that similar bursting is present when only considering nonstimulus periods. Bursting neurons were characterized by a peak in the low milliseconds range in the autocorrelogram (Fig 1B) and interspike interval (ISI) histogram. We used the peak of the log-transformed ISI histogram (Fig 1C) as in Nowak and colleagues [33] because this better distinguishes bursting behavior from high-rate Poisson-like behavior (see Materials and methods). Bursts typically consisted of 2 to 3 spikes (in agreement with those found for primate cortical bursting neurons [9,27]) and the distributions of the mean burst length for each unit is shown in S3 Fig. We also created 2 corroborating features: (1) an autocorrelogram metric that measures the relative height of the autocorrelogram at short and long time lag; and (2) the logISIdrop that detects a sharp drop from a short interval peak in the log(ISI) histogram. Units were classified as bursting ("Bu") if they had (1) ISI peak <10 ms; (2) autocorrelogram metric >0.5; and (3) logISIdrop > 0.2. If the unit fulfilled 2 of the 3 criteria, it was not classified, and otherwise it was classified as nonbursting. Bursting and nonbursting units are separated in scatterplots and provide a compelling split of the peak ISI times (Fig 2A and 2B, marginal histogram).

We confirmed that bursting was present not only in response to stimuli, but also throughout the neural activity. It was evident from inspection that bursts occurred during the pre and poststimulus time. For an example unit (S1 Fig), we show expanded burst occurrences during the unstimulated period, as well as autocorrelograms calculated from the entire protocol, from the pooled prestimulus periods, or from a longer segment of spontaneous activity. All 3 autocorrelograms indicated bursting with similar properties. As our protocols did not typically contain long periods of unstimulated baseline, we calculated the logISIdrop (which only relies on ISI below 16 ms) on segments of prestimulus time (200 ms) pooled across stimuli, repetitions, and protocols. The logISIdrop calculated from the entire recording and from pooled prestimulus time were highly correlated (Fig 2C; $r = 0.88$). Units where prestimulus logISIdrop exceeded 0.3 were labeled as "PBu," and bursting status determined in this way was in agreement with the "Bu" designation for 375/381 units. Similarly, in Katai and colleagues [27] and Onorato and colleagues [6], the bursting propensity and characteristics were similar between stimulated and unstimulated periods, suggesting that it is neuronal type property.

Next, we examined the population of nonbursting units. Barthó and colleagues [34] found trough-to-peak time ($t_{TTP}$) of the broadband (1 Hz to 5 kHz) extracellular waveform to be most informative for RS/FS separation. We acquired our signals filtered at 1 Hz to 5 kHz and only applied broad zero-phase digital filters to minimally distort waveform features [35–38].

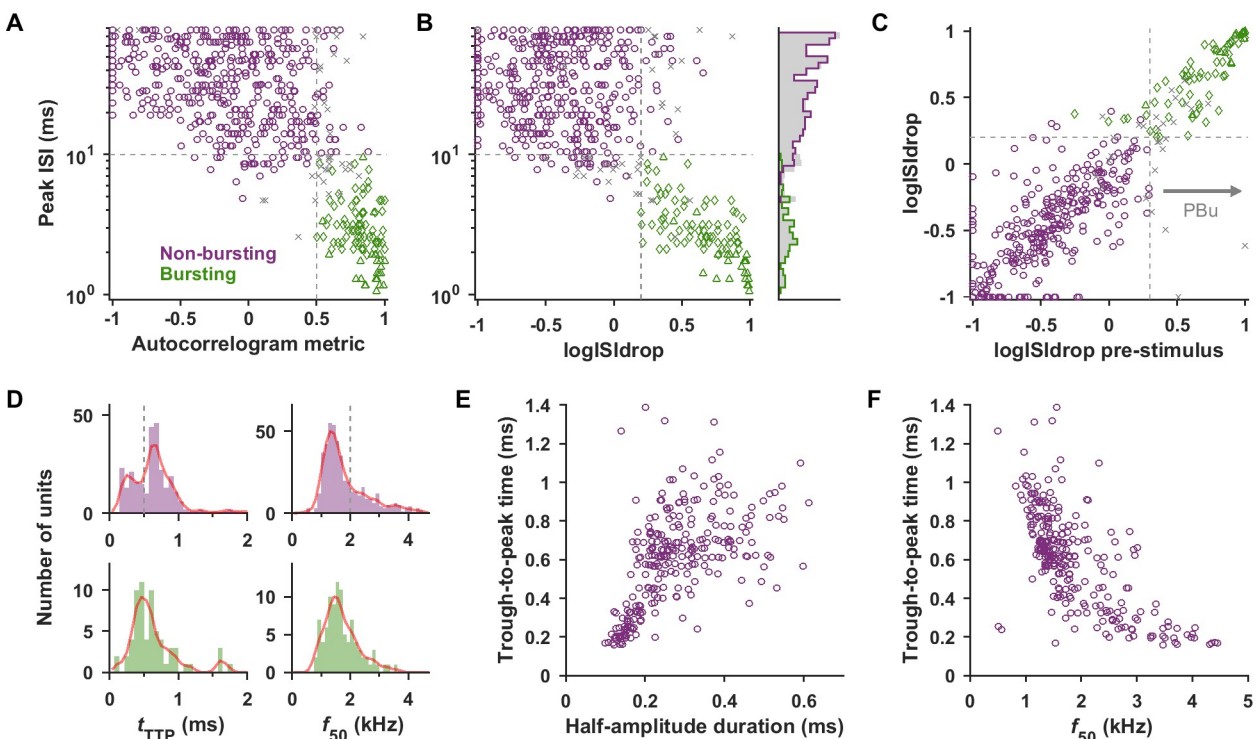

**Fig 2. Bursting units could be separated from nonbursting units, and nonbursting units could be divided into RS- and FS-like populations.**
(A) Scatter plot of the peak ISI versus the autocorrelogram metric (a proxy for relative amount of short time scale autocorrelation). Color indicates label assigned by consensus criteria-based classification (boundaries shown in gray dashed lines, ambiguous unclassified units as gray x's). The Bu1 and Bu2 subtypes of bursters (described in a later section) are distinguished as triangles and diamonds, respectively. (B) Scatter plot of peak ISI versus the logISIdrop metric. Marginal histograms are shown on the right for each labeled population (colored lines) and for the entire population including unclassified units (shaded gray bars). (C) Bursting occurs during spontaneous activity as well as the stimulus-driven period (see also S1 Fig). Comparison of logISIdrop calculated from the entire recording versus from the prestimulus period alone showed the 2 were strongly related. The group of units labeled as bursting based solely on having prestimulus logISIdrop > 0.3 are referred to as "PBu." (D) Histograms and kernel density estimates (red lines) of $t_{TTP}$ and $f_{50}$ values for nonbursting units (purple) and bursting units (green). $t_{TTP}$ for nonbursting units shows bimodality with groups above and below approximately 0.5 ms (Hartigans' dip test $p < 0.001$), presumably corresponding to RS and FS units. $t_{TTP}$ for bursting units is intermediate and peaks around 0.5 ms. $f_{50}$ for nonbursting units is also suggestive of bimodality with 2 overlapping groups— one centered around 1.5 kHz (RS) and one extending out above 2 kHz (FS), but Hartigans' dip test did not reach significance. Boundaries for criteria-based classification of nonbursting units into RS and FS are shown (gray dashed lines). (E) Scatter plot of nonbursting unit $t_{TTP}$ versus half-amplitude spike duration shows a broad cluster above $t_{TTP}$ = 0.5 ms and an elongated group below. The distribution is similar to that seen in Barthó and colleagues [34], where inhibitory/excitatory status was confirmed by crosscorrelogram analysis. Eight outliers with $t_{TTP} > 1.6$ are beyond the axes of the plot. A random jitter of up to half the minimum resolution was added to offset points with the same coordinates. (F) Scatter plot of $t_{TTP}$ versus $f_{50}$. Units with $t_{TTP}$ below approximately 0.5 ms generally had $f_{50}$ above 2 kHz. Data underlying this figure can be found in S1 Data. FS, fast-spiking; ISI, interspike interval; RS, regular-spiking.

For the nonbursting population (presumed to include RS and FS units), $t_{TTP}$ was bimodal (Fig 2D, Hartigans' dip test $p < 0.001$). $t_{TTP}$ for bursting neurons peaked around 0.5 ms (Fig 2D), intermediate between the putative RS and FS populations. Therefore, if bursting units were not first removed, the resulting spike width histogram may not appear bimodal. Similarly, in Trainito and colleagues [7], where cortical neurons were split into 4 groups, the authors noted that individual features were not necessarily bimodal, but that the use of multiple features allowed for separation of groups that were less distinct in 1 dimension. Similarly, their intermediate-waveform bursting group was split up if only 2 clusters were allowed. Therefore, it is important to use multiple criteria or features to tease apart these multiple overlapping groups. The scatter plot of $t_{TTP}$ versus spike half-amplitude duration for nonbursting units (Fig 2E) is similar to that shown in Barthó and colleagues [34], where a $t_{TTP}$ of approximately 0.5 ms divided an elongated cloud of narrow waveforms from a large cluster of broad waveforms.

To corroborate $t_{\mathrm{TTP}}$ measurements, we added a spectral measure of spike width. We computed the baseline-subtracted frequency spectrum of the average spike (e.g., Fig 1D, right of each pair). The high frequency at which the amplitude rolled off to 50% of the peak was termed the $f_{50}$. As expected, $f_{50}$ and $t_{\mathrm{TTP}}$ were inversely related, with $t_{\mathrm{TTP}}$ greater than 0.5 ms roughly corresponding to $f_{50}$ less than 2 kHz (Fig 2F). Nonbursting units were labeled "FS" if they satisfied at least 2 of 3 criteria: (1) $t_{\mathrm{TTP}} < 0.5$ ms; (2) $f_{50} > 2$ kHz; and (3) spontaneous rate > 5 spk/s; were labeled "RS" if they satisfied at least 2 of the following: (1) $t_{\mathrm{TTP}} > 0.5$ ms; (2) $f_{50} < 2$ kHz; and (3) spontaneous rate < 3 spk/s; and were otherwise not classified.

The various unit types were interspersed among each other and distributed throughout the topographical map of the recording area (S2F and S2G Fig). They also did not differ markedly in best frequency or recording depth (S2B and S2C Fig), although overall our recordings were biased toward superficial layers due to the superficial approach and long recording times. Multichannel or laminar probes could be used to more precisely establish laminar distributions of unit types. Despite having grossly similar distributions in location, depth, and best frequency, classified types differed in terms of a number of basic properties (S3 Fig). The first 6 displayed properties were used in the criteria-based classification process, and not surprisingly differed between the groups, but other properties differed as well. Both $t_{\mathrm{TTP}}$ and $f_{50}$ suggest that bursting unit spike width was intermediate between RS and FS units, in agreement with other studies [7,22]. FS units were characterized by high spontaneous and driven rates and unimodal ISI distributions after log transform (nonsignificant Hartigans' dip test $p$-values). For bursting units, log(ISI) was typically not unimodal due to overrepresentation of short intervals, and the percent of ISI less than 5 ms (a criteria often used to identify bursting units) was much higher than for RS and FS units, especially when normalized to that expected for a Poisson process with the same overall rate. Differences in latency and refractory period were also seen.

## Clustering-based categorization of units

To support our criteria-based classification, we tested whether a clustering method would detect the same classes. As noted in Trainito and colleagues [7], the use of multiple features allows for better discrimination of latent components. We selected 8 features that were differentially distributed between unit types and available for the largest number of units (see Materials and methods). Beyond the selection of these features, the method should yield an objective classification. We first standardized the data and performed dimensionality reduction by principal component analysis (PCA). The loading plot shows 2 main groups of correlated (or anticorrelated) features, namely those pertaining to burstiness and those pertaining to FS versus RS (Fig 3A). We then fit a Gaussian mixture model (GMM) to the first 3 principal components, which account for approximately 84% of the variance (Fig 3C). GMMs can accommodate components that are overlapping or elongated. The model indicates 3 primary clusters, based on the Akaike information criterion (AIC) and Bayesian information criterion (BIC) (Fig 3D), as well as the negative log-likelihood of cross-validation (Fig 3E). The contours of the 3 fit clusters correspond well with the 3 populations previously labeled by criteria (Fig 3B). For each unit, the GMM generated a posterior probability of it belonging to each type; units where the probability of belonging to 1 type exceeded twice the probability of either of the other 2 types were assigned to the most probable type. The GMM was in agreement with the criteria-based classification for approximately 94% of units labeled by both methods and the comparison lay mostly on the diagonal of the confusion matrix (Fig 3F). A total of 289 units were classified as RS, 144 as FS, 102 as bursting, and 46 did not fulfill the criteria for confident classification. More RS units may have been lost due to ambiguity at the border with both the FS and Bu clusters. The proportions of unit types from the criteria and GMM

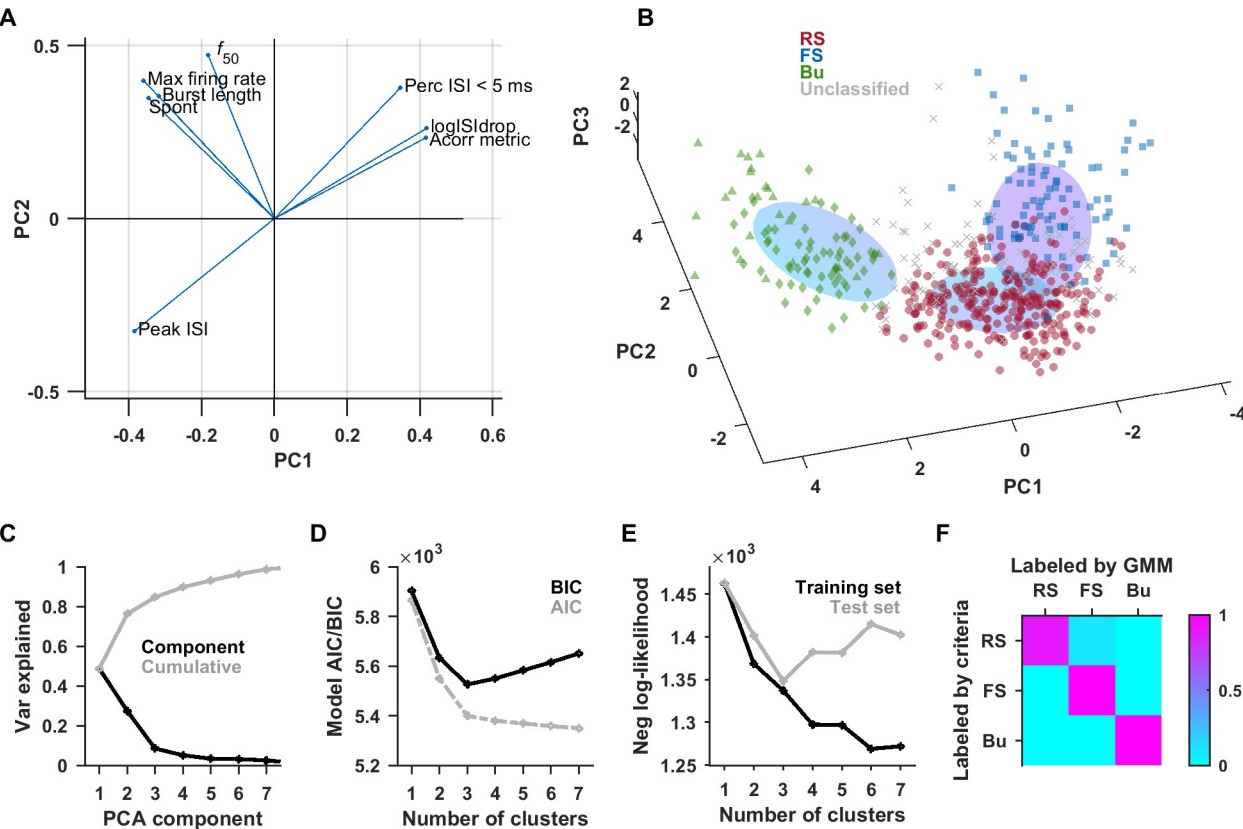

**Fig 3. Unsupervised classification also detects 3 major classes with strong agreement with criteria-based classification.** PCA was performed on a set of 8 features. The loading plot (A) revealed 2 main subsets of features, those pertaining to spike timing and burstiness, and those pertaining to fast spiking. We fit a 3-component GMM to the first 3 PCA components (B). The color of the points indicates the labels assigned by the previous method using criteria (triangular and diamond markers indicate Bu1 and Bu2 subgroups, respectively). The elongated ovoids represent the 3D contours of each of the 3 components of the GMM at half height. (D) The plot of the AIC and BIC versus number of components both support the choice of 3 components. (E) The data set was split into training and test sets (repeated 20 times), and the negative log-likelihood was calculated as a proxy for quality of fit. The negative log-likelihood increased after 3 components for the test set, suggesting that more than 3 components resulted in overfitting. (F) The confusion matrix between the type labels assigned by criteria and GMM showed high agreement. Data underlying this figure can be found in S1 Data. AIC, Akaike information criterion; BIC, Bayesian information criterion; GMM, Gaussian mixture model; PCA, principal component analysis.

methods are in rough agreement with studies in other parts of cortex [6,7,27,39], but the exact proportions can vary by cortical area [7].

## Unit type–specific differences in temporal coding properties

In a subset of units, functional responses were assessed in detail. Acoustic stimuli were optimized in frequency, sound level, and bandwidth for each unit, as relatively optimal stimuli drive the most sustained responses [32]. In the first experiment, units were presented with synthetic stimuli ranging logarithmically in duration from 12.5 to 400 ms. Bursting units showed the strongest adaptation, while FS units showed the most sustained responses (Fig 4A). We noticed that a subset of bursting units fired only at onset and had almost no sustained activity. Bursting units with intraburst frequency above 500 Hz were almost exclusively of this type. When bursting units were separated based on intraburst frequency into Bu1 ($>$500 Hz) and Bu2 ($\leq$500 Hz) subgroups, the mean Bu1 response was strongly adapting (Fig 4A). We then quantified adaptation using an adaptation index for individual units that compared the rate in the first 100 ms versus last 100 ms of the response window for 200

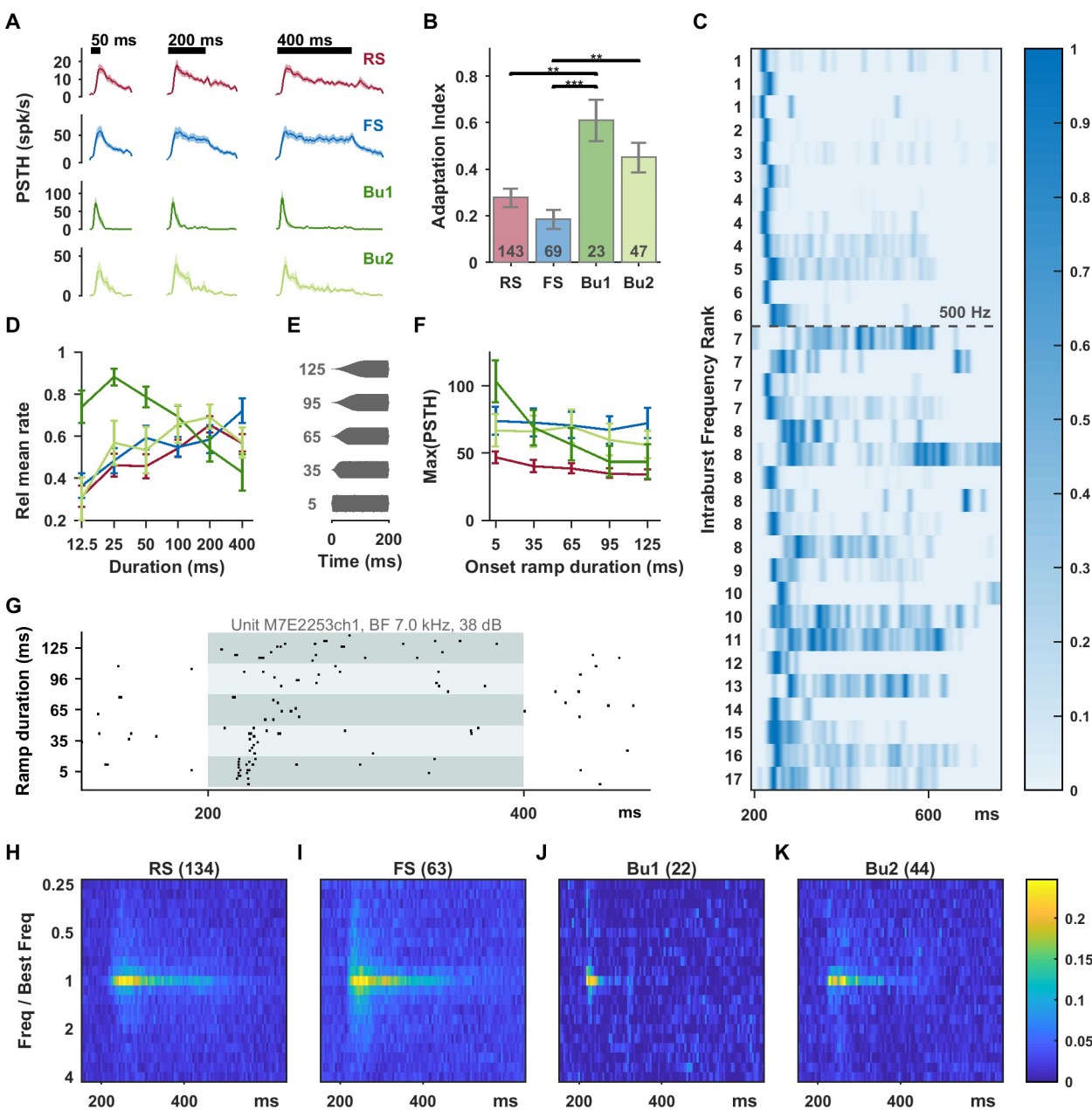

**Fig 4. Bursting units had the most strongly adapting responses to sustained stimuli at best frequency and were sensitive to the rate of sound onset.** (A) Smoothed PSTHs in response to 12.5, 50, and 400 ms stimuli at best frequency, bandwidth, and sound level. Bursting units had rapidly rising and strongly adapting onset responses. The effect is clearest for the Bu1 subgroup (intraburst frequency >500 Hz). Shaded bands shown standard errors; $n$ values are RS (78), FS (44), Bu1 (12), and Bu2 (20). (B) An adaptation index was calculated from responses to 200 ms best frequency stimuli (larger values indicate more adaptation). Welch's ANOVA revealed a significant group difference ($F_{3,76.6} = 9.2$, $p < 0.0005$) and significant pairs from the Games–Howell post hoc test are shown. Again, bursting units showed the most adaptation, and FS units the least. Error bars show SEM. (C) Max-normalized smoothed PSTH's (400 ms stimuli) for bursting units sorted from highest to lowest intraburst frequency (units with same frequency assigned equal rank). Bu1 units (>500 Hz) appear above the dashed line. (D) Driven rate, calculated over the entire response window and normalized to the maximum, peaks at 25 ms for Bu1 (due to lack of sustained response). In response to stimuli with increasingly slow onset ramps (E), Bu1 unit spiking became more distributed (example shown in (G)) or even nonresponsive. Maximum PSTH height decreased substantially with slower stimulus onsets for Bu1 units, but not for the other types (F). $n$ values for ramp rate are RS (57), FS (29), Bu1 (11), and Bu2 (16). Recentered response heatmaps in (H), (I), (J), and (K) show temporal and receptive field differences between the unit types. Data underlying this figure can be found in S1 Data. PSTH, peristimulus time histogram; RS, regular-spiking.

ms standard stimuli. This index ranged from −1 to 1, with 0 indicating no adaptation and 1 indicating complete adaptation by the last 100 ms (Fig 4B). Again, bursting units were significantly more adapting than RS and FS units, with the largest effect for the Bu1 group. Recentered receptive fields from units well driven by the standard tuning protocol were averaged by unit type and show the brevity of Bu1 responses, among other differences (Fig 4H–4K). RS units had a particularly arc-shaped response onset, with latencies for nonoptimal frequencies being significantly delayed relative to latencies for the best frequency, possibly reflecting longer integration time to spiking. When viewing the responses to 400 ms long stimuli sorted from highest to lowest intraburst frequency (Fig 4C), the Bu1 group (above the dashed line) consisted predominantly of units with precise onset responses, whereas the Bu2 group includes many sustained responses. Bu1 units also had narrower spikes and shorter latencies and refractory periods than Bu2 units (S3 Fig). Driven rate averaged over the response window decreased substantially with duration above 25 ms for the Bu1 group (due to a lack of sustained response) (Fig 4D).

Bu1 responses also showed high sensitivity to the rate of sound onset for ramped stimuli (Fig 4E–4G). Bu1 units responded to fast onsets with precisely timed bursts, but responded in a distributed way or not at all to slower onsets (example in Fig 4G). The peak peristimulus time histogram (PSTH) rate decreased monotonically with stimulus onset rate for Bu1 units (Fig 4F). Although Bu1 responses were precise and transient to fast onset stimuli, they were not "binary spiking" [40] (Fig 4G, Fig 6 insets).

Bu1 units showed related properties when stimulated with sinusoidal amplitude modulation (SAM) at 2 to 512 Hz (logarithmically spaced, 100% modulation depth, carriers at best frequency, bandwidth, and level, or 30 dB above threshold for monotonic units). We calculated the vector strength (VS) as a measure of the tendency for spikes to occur at a particular phase of the modulation (phase-locking). Similarly to Bendor and Wang [41], nonsignificant (spurious) VS values were set to 0 (see Materials and methods). The first 50 ms after stimulus onset were not included such that a pure onset response would not generate a high VS. The RS and FS groups had lower VS and more nonsignificant units as compared to the bursting groups (Fig 5A and 5B). The averaged VS profile of Bu1 units has a bandpass shape peaking at 8 to 16 Hz, consistent with preference for more rapidly modulated stimuli with higher onset slopes. The maximum modulation rate that a unit could synchronize to was also highest for the Bu1 group (Fig 5C). A few FS units were also significantly synchronized to high modulation rates, which could be due to a subpopulation not well separated by the current classification or the large number of spikes in FS units making it easier to achieving significance on the Rayleigh statistic.

Marmoset auditory cortex neurons have long been classified as either synchronized or nonsynchronized in their responses to SAM or click trains [10,11]. The Bu1 population had the largest fraction of units with synchronized responses above 4 Hz (Fig 5D). The difference was even more pronounced for synchronization at 16 Hz or above (Fig 5E): Only about 20% of RS units but almost all Bu1 units were synchronized. Among synchronized units, Bu1 units had a mean synchrony-based best modulation frequency (tBMF) of 15.9 ± 3.3 Hz (SEM, maximum of 64 Hz), as compared with 5.6 ± 0.4 Hz for RS, 7.5 ± 1.2 for FS, and 6.7 ± 1.3 for Bu2 units. In the average period histograms for 2 Hz SAM, RS and FS units generally followed the shape of the stimulus, while bursting units peaked early in the cycle during the rising phase (Fig 5F). This slope-triggered behavior to sinusoidally varying input has been previously described for bursting neuron models [28]. Similar results were seen for the groups classified by the GMM, and bursting neurons as a unified group had similar behavior whether classified by criteria, prestimulus logISIdrop, or the GMM (S5 Fig). A few neurons in our sample had nonsynchronous and sustained responses that were narrowly rate-tuned to particular high modulation

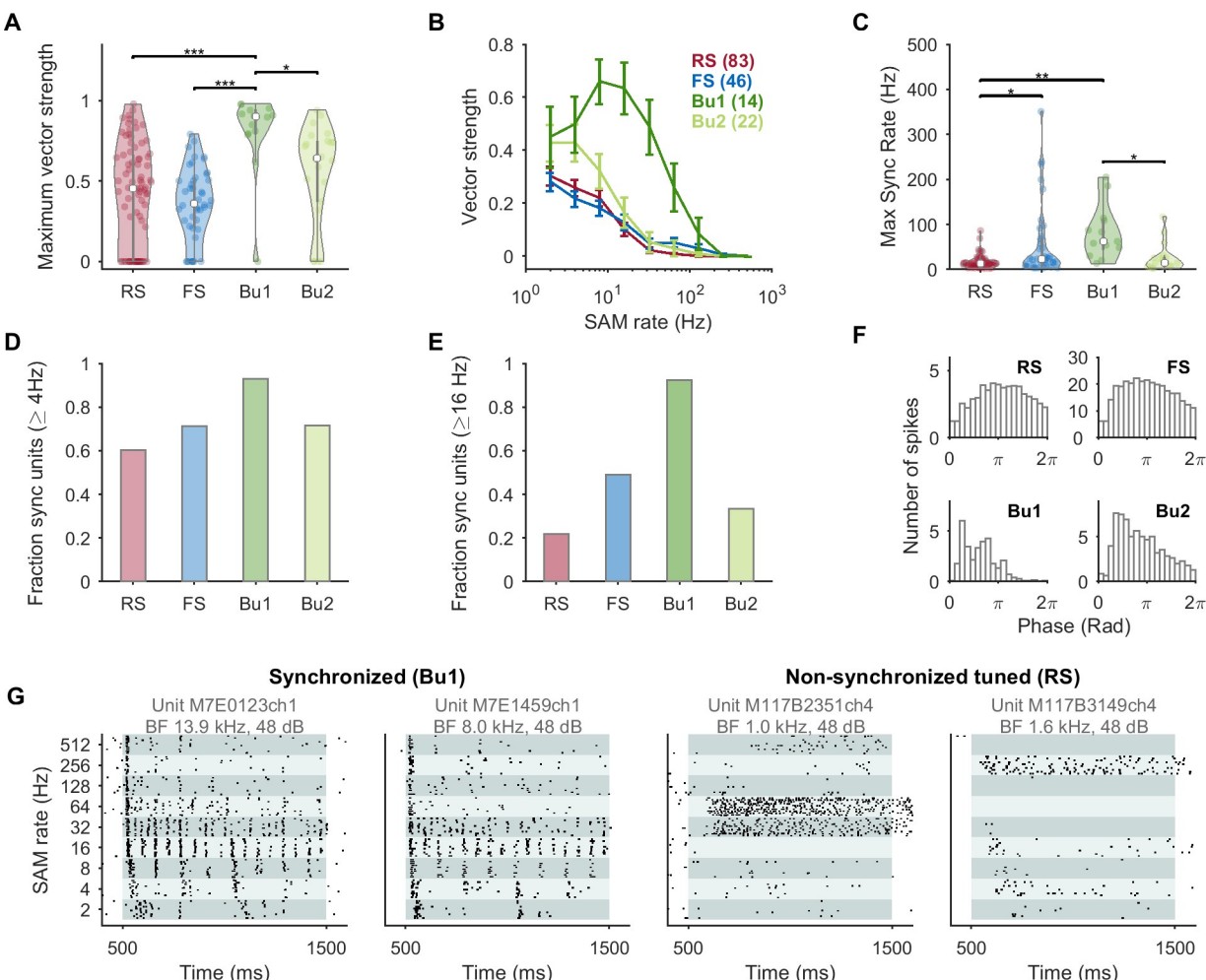

**Fig 5. In response to sinusoidally amplitude modulated sounds, bursting neurons had higher VS, were more likely to be synchronized and up to higher rates, and responded in the early phase of the cycle.** (A) Violin plot of maximum VS. Nonsignificant VS values were set to 0 (see Materials and methods). Bursting types had high VS values. For (A) and (C), Welch's ANOVA was statistically significant ($F_{3,42.0}$ = 11.3, $p < 0.0005$) and significant pairs from post hoc testing are indicated. (B) Mean VS versus modulation rate. Bu1 units show bandpass behavior; Bu2 also has elevated VS. (C) The maximum modulation rate with a significant VS was much higher for the Bu1 population, although some FS-like units were also able to phase-lock to high modulation rates. Units were considered to be synchronized if they had a significant Rayleigh statistic at or above 4 Hz (D) or 16 Hz (E) and a VS of at least 0.1; units with no significant temporal or rate response in the modulation range considered were excluded. A majority of Bu1 units were synchronized. (F) Average period histograms for stimulation at 2 Hz. (G) Examples of 2 highly synchronized units labeled Bu1 by criteria and Bu by the GMM (left) and 2 nonsynchronized units that were tuned to SAM rate that were labeled RS (right). Data underlying this figure can be found in S1 Data. Bu, bursting; FS, fast-spiking; GMM, Gaussian mixture model; RS, regular-spiking; SAM, sinusoidal amplitude modulation; VS, vector strength.

rates (Fig 5G) and were reminiscent of examples shown in previous studies (e.g., [11]). These units may be underrepresented as they are not well driven by unmodulated stimuli. In our sample, such units were nonbursting and contrast with the 2 highly synchronized example Bu1 units.

To confirm that burstiness in Bu1 units was not due solely to their burst-like onset responses, we compared the logISIdrop calculated from pooled prestimulus time for these units. In 20/25 Bu1 units, the logISIdrop could be calculated on pooled prestimulus time (having at least 50 total ISI values), and in every case it was above 0.2 (our threshold for bursting), with a mean value of 0.86 ± 0.04, indicating spontaneous burstiness. The Bu1 and Bu2

subgroups could also be derived by directing a GMM to create 2 groups using the features intraburst frequency, logISIdrop, and latency on the population of bursting units. The adaptation, onset slope-sensitivity, and phase-locking properties of the Bu1 and Bu2 groups from this clustering were similar to those created by the 500 Hz criteria on intraburst frequency (S4A–S4D Fig).

In the last set of experiments, units were presented with examples of 10 marmoset calls and their time-reversed counterparts ("Mixed Vocalizations List," S6A Fig). In some cases, we also presented lists consisting of only 1 particular call type ("Call Type Lists," S6B and S6C Fig). For subsequent quantitative analyses, the standard "Mixed Vocalizations List" was used (and raster plots can be seen in Figs 1E and 7D as well as S7 Fig), but raster plots for 2 "Call Type Lists" are shown in Fig 6 to illustrate unit type differences when comparing the same call types. The call types varied in their temporal modulation content: *phee* calls typically only have 1 onset, while *trill* calls have a frequency and amplitude modulation at approximately 30 Hz. Yet even for the same call type, both precise transient and imprecise sustained responses could be observed and these correlated with unit type (Fig 6). The bursting units shown had intraburst frequencies of 476 Hz or higher, responded mostly at the onset of the *phee* call, and phase-locked to the *trill* call. RS units had more diffuse rate responses, while FS units were excited or inhibited in a slow manner for the *phee* call and more rapidly for the *trill* call. Examples of bursting unit responses to the "Mixed Vocalizations List" also showed strikingly temporally precise but diverse responses (S7 Fig).

To quantify these differences, we identified the stimuli that each unit was responsive to and computed the correlation index (CI) [42], which can be seen as an extension of VS to aperiodic stimuli, as an indicator of precise spiking across repetitions (see Materials and methods, S8 Fig). Responsiveness was determined either based on a significant increase in firing rate over the entire stimulus, or in individual 5 ms time bins. RS units had the highest vocalization selectivity, while Bu1 units responded to more stimuli when considering instantaneous responses rather than overall rate (Fig 7A). Note that units could also be suppressed by vocalizations, which would contribute additional contrast for decoding. The maximum CI value across stimuli with excitatory responses in the "Mixed Vocalizations List" was termed the $CI_{max}$. FS units had the lowest $CI_{max}$ values, RS units were intermediate, and bursting units (particularly Bu1) had the highest $CI_{max}$ values (Fig 7B). This pattern was significant regardless of whether bursting was identified using criteria, prestimulus logISIdrop, or the GMM classification. When ranked from highest to lowest $CI_{max}$, the Bu1 population had the highest rankings (all within the top 20%), followed by Bu2, RS, and then FS (Fig 7C). A Kolmogorov–Smirnov test between RS and Bu1 units rejected the null hypothesis that the 2 samples came from the same distribution ($p < 0.001$). A similar result was seen between RS and all bursting units combined ($p < 0.001$). When the topographical locations of units were projected onto the lateral sulcus, the primary axis of regional variation in this study, the $CI_{max}$ of bursting units and in particular Bu1 units was consistently higher than that of other units at the same position along the axis (S2H and S2I Fig). Therefore, the robust unit type differences we saw are unlikely to be accounted for by confounding associations with region, depth, or best frequency (S2 Fig).

To test whether these differences impact stimulus decoding, we used the Victor–Purpura spike distance metric (43) to classify response spike trains to the stimulus class with which it had the shortest average distance (with power transformation). The transmitted information (*H)* is plotted as a measure of decoding performance as a cost parameter (*q*) is varied, spanning from rate coding (low *q*) to precise temporal coding (high *q*). For the illustrative units in Fig 7D, the precise fast bursting units were right shifted (toward temporal coding) relative to the more imprecise RS units (Fig 7E). This pattern was seen in the averages by unit type (Fig 7F and 7G). At low *q*, RS units performed better, while at high *q*, Bu1 units performed better. The

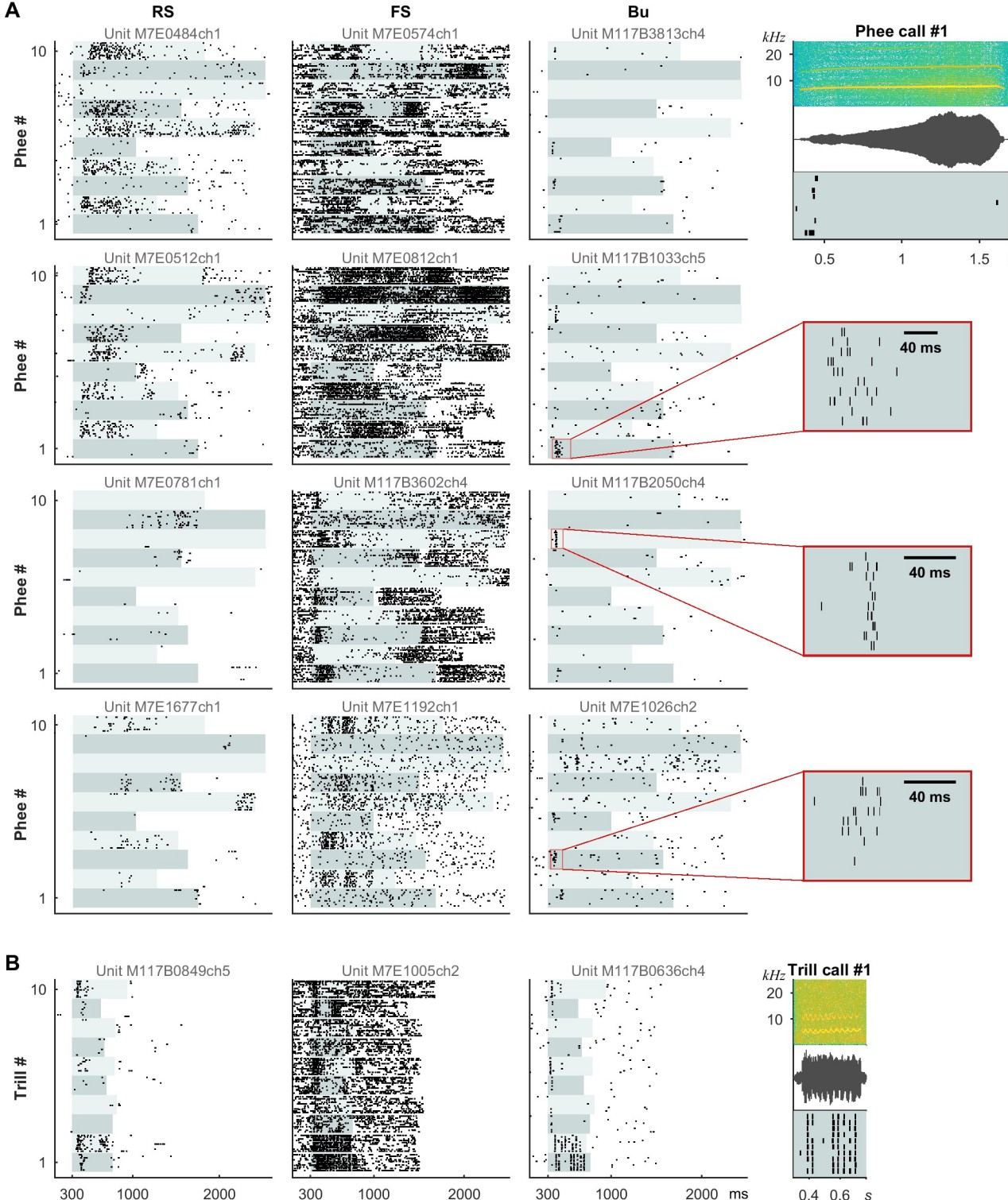

**Fig 6. RS, FS, and Bu units had distinct types of responses to vocalizations analogous to their responses to sustained and SAM stimuli.** Responses are shown to "Call Type Lists" consisting of example tokens of the sustained *phee call* or the rapidly fluctuating *trill call*. Example spectrograms, waveforms, and corresponding response raster plots for bursting units are shown on the right, while complete stimulus spectrograms can be seen in S6B and S6C Fig. (A) Columns show responses of 4 example units of each type to 10 prerecorded *phees*. Bu units responded at particular moments, often at the onset, while RS and FS units were excited or inhibited throughout the call. (B) A Bu unit phase-locked to the rapid modulation (approximately 30 Hz) in some examples of *trill* calls. An FS unit was also capable of representing the modulation, but with many more spikes per

cycle. Units shown were consistently labeled by criteria and GMM. The 5 distinct Bu units had intraburst frequencies of 476, 476, 526, 588, and 909 Hz (top to bottom). Alternating light aqua shading indicates stimulus duration. Bu, bursting; FS, fast-spiking; GMM, Gaussian mixture model; RS, regular-spiking; SAM, sinusoidal amplitude modulation.

peak for FS units was also right shifted relative to RS units, suggesting that FS units can carry some rapid temporal information, but performance falls off steeply at very high $q$ values. Bu1 units, on the other hand, maintained high relative decoding performance up to very high values of $q$. The Bu1 and Bu2 subgroups clustered via GMM showed similar CI and decoding properties as those created by the 500 Hz intraburst frequency criteria (S4E and S4F Fig).

A previous study in awake macaque auditory cortex found a population of nonselective onset units (termed "stereotyped neurons") that were postulated to provide a temporal reference frame for other neurons [44]. However, our bursting units had diverse and selective responses to vocalizations (Figs 6, 7C and S7 Fig) and defined (although sometimes broad) tonal receptive fields (Fig 4H–4K). In visual cortex, bursting units were also described as being stimulus selective, even more so than FS units [6]. Even if onset units are selective, they could still function as a temporal reference frame to enhance decoding as demonstrated in Brasselet [44] and Hamilton [17].

To better understand the differential contributions of Bu and RS units to stimulus representation at the population level, we implemented a population decoder based on only bursting units, only RS units, or a mixture of both types. FS units were not included as they are presumed to be interneurons that do not project to downstream areas. The decoder predicted stimulus identity from population responses to the "Mixed Vocalizations List" (see S6A Fig) represented in 10 ms time bins. A leave-one-out design was used to assess performance on a left-out trial when trained on the remaining trials, as in [45]. The neural pool consisted of units responsive to at least 1 stimulus in the "Mixed Vocalizations List," and included only bursting units, only RS units, or a mixture of the 2 (maintaining the relative prevalence seen in our data). For each population size, a subset of units was sampled from the pool, and a different trial was randomly selected for each unit as the left-out test trial. This sampling procedure was repeated 50 times to generate a confusion matrix. Population sizes were chosen to make use of all 54 available bursting units and 110 RS units.

We assessed the performance of both simple maximum correlation coefficient (MCC) decoding and linear discriminant analysis (LDA) decoding (see Materials and methods). The 2 approaches performed well and very similarly to each other. For a population of 54 bursting units, accuracy was 0.98 for MCC and 0.97 for LDA. For a population of 54 RS units, accuracy was only 0.67 for both MCC and LDA. Confusion matrices showed clear diagonals of correct classification, but incorrect classifications were much more common for the RS population (Fig 8A). For the subsequent panels, MCC was used as it was much faster for large feature sets. Although decoding accuracy generally improved with population size, bursting units performed substantially better than RS units for a given population size and the mixture performed intermediately (Fig 8B).

To test the impact of removing temporal information, we averaged the response of each unit across time bins to produce an average rate code (Fig 8C, "Avg time"). To test the impact of removing unit identity, we pooled across all units while maintaining temporal information (Fig 8C, "Avg units"). Both manipulations severely impaired decoding with 54 units of either RS or Bu units, suggesting that both types of units contain information across the population as well as in their temporal response patterns. However, Bu units were more strongly impaired for loss of temporal information than loss of unit identity and still performed reasonably well when using only the mean population response over time.

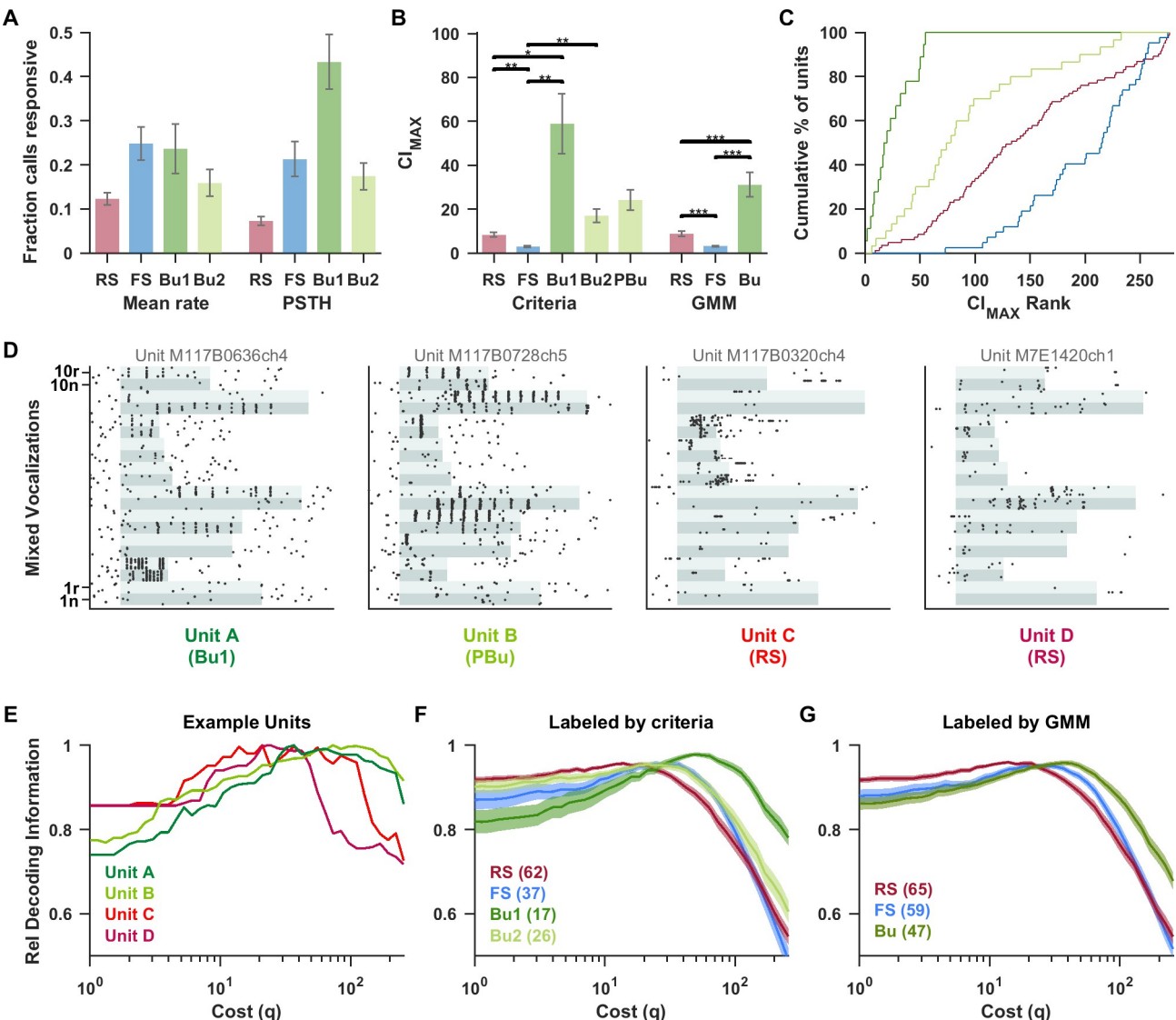

**Fig 7. High-frequency bursting neurons spiked in a temporally precise manner in response to vocalizations and better supported a temporal code while RS neurons showed higher rate selectivity.** Here, we used responses to a standard "Mixed Vocalizations List" of 20 simple and compound calls (spectrograms shown in S6A Fig). (A) The fraction of call stimuli for which units had an excitatory response (as assessed by overall firing rate or for 5 ms PSTH bins (see Materials and methods). RS units had the highest vocalization selectivity. Bu1 units responded to more stimuli in terms of instantaneous rate change than overall driven rate, reflecting phase-locking. *n* values are RS (158), FS (50), Bu1 (18), and Bu2 (44). (B) For each unit, the maximum CI across responsive stimuli in the "Mixed Vocalizations List" was considered the $CI_{max}$. Bu units had the highest $CI_{max}$ values, indicating that they spiked at particular times during the stimulus. The groups ("RS," "FS," "Bu1," and "Bu2") as labeled by criteria and ("RS," "FS," and "Bu") as labeled by the GMM were each compared with Welch's ANOVA and found to be highly significant ($F_{3,48.9} = 12.5$, $p < 0.0005$ and $F_{2,90.5} = 22.5$, $p < 0.0005$). "PBu" was not compared. *n* values for criteria-based categories are: RS (83), FS (42), Bu1 (18), Bu2 (30), PBu (58); and for GMM-based categories are: RS (86), FS (68), and Bu (54). (C) Units were ranked from highest to lowest $CI_{max}$. The cumulative histogram shows that all Bu1 units had high-ranking $CI_{max}$ values, followed by Bu2, RS, then FS groups. (D) Examples of the vocalization responses of 2 bursting units (intraburst rates 909 and 667 Hz) and 2 RS units to the "Mixed Vocalizations List" (S6A Fig) of natural ("n") and time-reversed ("r") vocalization tokens. The stimulus duration is indicated with alternating aqua shading, with each band corresponding to all 10 repetitions of the given stimulus (stimuli were presented in interleaved order). We used the Victor–Purpura distance between spike trains [43] to decode the stimulus from individual trains. A cost parameter (*q*) was varied, spanning from rate coding to temporal coding as *q* is increased. (E) For the example units in (D), we calculated the transmitted information (*H*) of the confusion matrix as a measure of decoding quality. To better compare the effect of changing *q*, *H* is normalized by its maximum value for each unit. The 2 precise bursting units had a higher optimal *q* range than the RS units. At *q* = 1, where decoding is close to a rate code, the RS units performed well relative to the bursting units. In contrast, at high *q* values, decoding performance of RS units falls off steeply while that of high-frequency Bu units persisted. Similar trends are seen for the average of all units labeled by criteria (F) and by the GMM (G). Data underlying this figure can be found in S1 Data. Bu, bursting; CI, correlation index; FS, fast-spiking; GMM, Gaussian mixture model; PSTH, peristimulus time histogram; RS, regular-spiking.

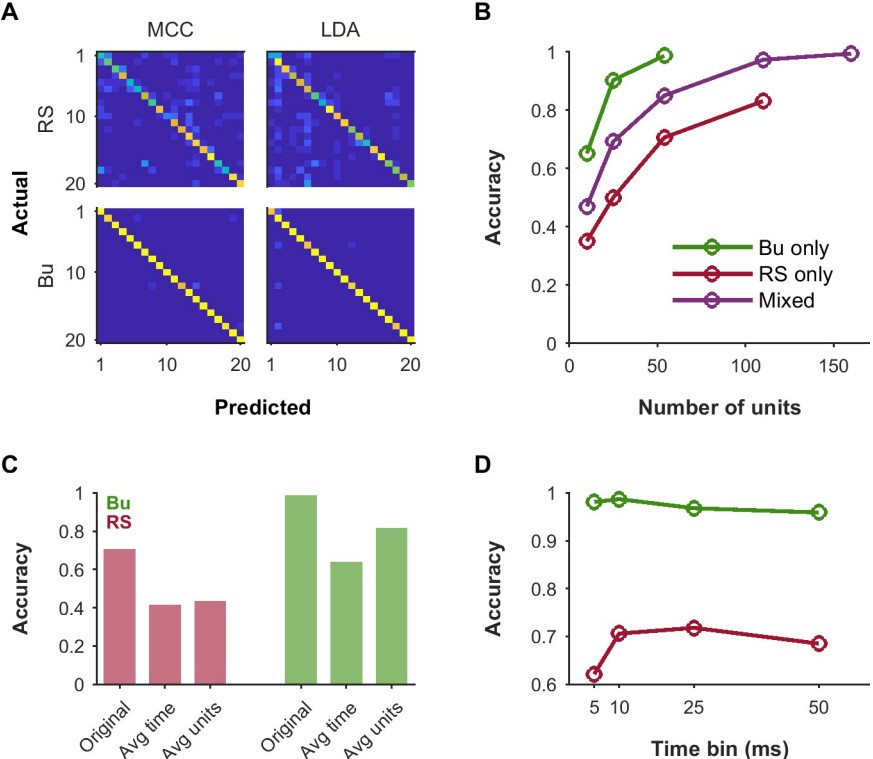

**Fig 8. Both RS and Bu units could contribute to population decoding, but Bu unit populations achieved particularly high accuracy with relatively few units.** To assess the contributions of RS and Bu unit types to downstream processing, we simulated population decoding for responses to the "Mixed Vocalizations List" (S6A Fig). (A) Confusion matrices illustrating decoding for populations consisting of only RS or only Bu, using an MCC decoder or LDA. The population size was 54 units (the maximum number of available Bu units). (B) Growth of accuracy with size of population used for decoding, for populations of Bu only, RS only, or a mixture of both types in the ratio naturally observed in our data. (C) Impact of collapsing across time bins ("Avg time") versus across units ("Avg units"). In the first case, we took the mean of the response over all time bins to produce an average rate code. In the latter case, we averaged across units while preserving time bins to produce a pooled temporal response. Performance was negatively impacted by each of these manipulations, but notably, Bu populations were more impacted by the loss of temporal information than the loss of unit identity. (D) Accuracy as a function of the size of the time bins used in decoding. While performance of RS populations suffered at the finest temporal resolution of 5 ms, Bu population performance was very good at 5 ms and optimal at around 10 ms, suggesting different informative timescales in these unit types. Data underlying this figure can be found in S1 Data. Bu, bursting; LDA, linear discriminant analysis; MCC, maximum correlation coefficient; RS, regular-spiking.

Lastly, we looked at the impact of using larger or smaller time bins (Fig 8D). Smaller time bins might be noisier, while larger time bins might obscure fine temporal information. The overall effect would result from a balance of these factors in relation to the actual jitter and modulation time scales present in the unit responses. Indeed, a roughly inverted-U-shaped behavior is observed. However, while RS units performed relatively poorly with 5 ms time bins and best with 25 ms time bins, Bu units were optimal at 10 ms and almost equally good with 5 ms time bins. Note that RS populations still supported less accurate decoding relative to Bu populations at all time resolutions tested.

## Discussion

We classified a majority of extracellular units using either a few chosen criteria or an unsupervised clustering method. The method of criteria gives more direct insight into the basis of the classification (primarily burstiness, spike width, and firing rates), while the unsupervised

method gives more objective support for the classification. The latter may generalize better to other recording setups, species, or parts of the brain. The optimal number of clusters was found to be 3, but more clusters may be identified with a larger sample or selection of other features. In Trainito and colleagues [7], 4 classes were found with the full data set of 2,488 units, but smaller subsamples of the data often produced fewer classes. We were not able to separate out a population proposed to consist of parvalbumin-negative inhibitory neurons with intermediate spike widths [7], but do see evidence for 2 subgroups of bursting units (Bu1 and Bu2).

The extent to which burstiness exists as a continuum versus as discrete types remains to be seen. However, our clustering method based on a sum of Gaussian distributions (with no restrictions on the covariance matrix) selected 3 primary clusters (Fig 3D and 3E); if all non-FS units were part of 1 elongated distribution, the model should have selected only 2 clusters. Furthermore, our bursting units plausibly correspond to biological neural types.

Traditionally, 2 types of bursting neurons have been described in cortex. Chattering or fast rhythmic bursting (FRB) neurons [25,33,46] have intraburst frequencies of 350 to 700 Hz, while intrinsic bursting (IB) neurons have intraburst frequencies <425 Hz [33]. Therefore, our Bu1 units (>500 Hz) may correspond to chattering-like neurons, while the Bu2 group may include a mixture. Chattering neurons have been described in higher mammals [21,33,46] and primates [19,24,39], but not in rodents, where bursting cortical neurons are IB-like [33,34,47,48]. Onorato and colleagues [6] described a sizeable population of chattering-like units in primate but not mouse V1 superficial layers. In primate frontal cortex, Katai and colleagues [27] found both chattering-like units with high-frequency bursts and IB-like units with lower frequency bursts (both groups were excitatory in cross-correlograms). Chattering neuron bursting relies on persistent $Na^+$ current rather than $Ca^{2+}$ current [21]. The presence of Kv3 channels that promote fast repolarization of the action potential in a subset of superficial layer non-GABAergic neurons in primate (but not mouse) may also play a role [49,50].

More recently, Patch-seq in human cortical tissue finds a category of glutamatergic neurons that are distinct in their gene expression, morphology, and electrophysiological features. These neurons fire bursts of action potentials at stimulus onset followed by strong adaptation [8]. They are speculated to correspond to superficial bursting pyramidal neurons observed in monkey cortex in [9] and may also correspond to our Bu1 population.

A number of studies, mostly in visual cortex, have documented a gamma frequency (30 to 80 Hz) rate of burst occurrence in chattering neurons during step current injection or optimal stimulus presentation [6,24–27,33,46]. However, burst timing is described as sporadic rather than oscillatory in area MT [19] and sensorimotor cortex [22]. Almost half of pyramidal neurons in primate dorsolateral prefrontal cortex layer 3 responded with bursts to current injection [9], but bursting occurred at onset, in contrast to the rhythmic bursting seen in visual cortical neurons [25]. Because our Bu1 units typically responded mostly at onset (Fig 4), autocorrelograms constructed from the stimulated period at best frequency were often very sparse. The raster plots did not show evidence of rhythmic bursting during sustained unmodulated stimuli. While log-transformation often produced bimodality in the ISI histogram, a prominent second peak was not typically seen in the ISI histogram without transformation nor in the autocorrelogram. Our Bu1 units appear different in this respect from FRB units in the visual system, possibly due to the high regional variability of pyramidal neurons in primates [51]. Evidence for stimulus-triggered gamma oscillations in auditory cortex is also equivocal. In Brosch and colleagues [52], sustained activity showed an increase in power above 41 Hz. However, Steinschneider and colleagues [53] separately analyzed the higher frequency bands and found the greatest increase in power unrelated to the evoked potential to be in the very high gamma band (correlated with spiking activity), with minimal change at 30 to 70 Hz. One

could speculate that prominent intrinsic oscillations in the gamma range would interfere with the auditory cortex's ability to process rapid temporal stimuli with their own time scales.

Our results reaffirm that neurons in awake marmoset auditory cortex typically fire multiple spikes when well driven [32], but suggest that in a particular subpopulation it may be advantageous to have a strongly adapting response. The use of duration protocols and a 200-ms standard stimulus helped dramatize the difference, which is more subtle at the shorter durations often used (Fig 4A). Bu1 units showed the strongest adaptation, highest VS, largest proportion of synchronized units, and most temporally precise vocalization responses. Roughly 5% of our classified units were labeled Bu1, but this is likely an underestimate since many units with intraburst frequencies below 500 Hz responded like Bu1 units. The majority of Bu1 units are very well synchronized, firing at a particular phase of the modulation, but other types of units could show synchronization as well. Our unit types bear resemblance to the early findings of de Ribaupierre and colleagues [54]: FS-like units showed entrainment but became sustained and unsynchronized above a limiting rate. One group of "regular-spiking" units showed responses limited to low pulse rates. Another group of "regular-spiking" units had precise latencies and phase-locking but became onset responsive above their limiting rate. Such a unit was noted as having an ISI peak near 0 representing double firing. Similarly, Lu and Wang [55] show highly synchronized example units with short-interval ISI peaks and hint at a relationship between bursting and synchronization, although it may have been presumed that these bursts were caused by periodic stimulation. Our results show that bursting is a unit type property present even in spontaneous activity.

Many functions have been proposed for burst firing throughout the nervous system, such as detecting coincident inputs, toggling between modes, simultaneously encoding multiple information streams with single and burst spikes, and triggering stronger synaptic release or plasticity [56]. Our results support temporal edge detection as another potential function of bursting. For slow 2 Hz SAM stimuli, bursting units responded during the positive slope of the modulation, as predicted by certain bursting neuron models [28]. Such bursting is mediated by positive feedback from persistent $Na^+$ current and terminated by negative feedback from slow-activating $K^+$ current. This may be a particular case of "class 3" excitability [57] whereby fast-activating inward current overpowers slow-activating outward current during depolarizing transients. This class also has low-threshold outward currents, which decrease the membrane time constant and further increase temporal precision [58,59]. The advantage of a bursting rather than single-spiking neuron of this sort include the ability for simultaneous graded encoding [28] and more reliable driving of the postsynaptic neuron [56,60,61]. This class of responses could be 1 cause of the observation of high temporal precision firing during dynamic stimuli but lower temporal precision during constant stimuli [62,63]. In the auditory cortex, synchronized neurons also showed higher precision in onset responses and to periodic or irregular events occurring at low or moderate rates [55]. Another example of this behavior is the octopus cell of the cochlear nucleus that is sensitive to the rate of depolarization and responds with high temporal precision to acoustic transients and periodic stimuli [58]. In weakly electric fish, bursting neurons report upstrokes (and by synaptic inversion, downstrokes) in the electric field signal [64]. Simple estimation methods did not provide a good description of these neurons and such behavior may cause inaccuracies in spectro-temporal response function (STRF) predictions.

Thus, intrinsic properties likely play a role in the functional behavior of the bursting neurons. The relatively nonadapting responses of FS units may also derive in part from their limited adaptation in response to current injections [33,48]. However, this does not preclude contributions from synaptic and circuit factors such as synaptic depression or delayed inhibition [54,65–67]. We do not imply that unit type explains all heterogeneity in temporal

responses—regional differences [68,69], laminar differences [70], and hemispheric differences [71,72] likely also play a role. Relative prevalence of bursting neurons can vary substantially between regions [7,9] and could contribute to regional differences in temporal encoding. Future research should clarify the relationship between neuronal type and these other factors.

Whether inherited and refined or created anew in the auditory cortex, the use of both rate and temporal coding can be seen in cortex [11,55] as well as earlier stages of auditory processing in the auditory nerve [73,74], cochlear nucleus [58,75], inferior colliculus [76], and thalamus [77]. Rate-based and temporal-based encoding may subsequently proceed in parallel streams from primary to secondary auditory cortical regions [17,69,78,79]. Populations specialized for transient versus sustained encoding also seem to be an organizing principle in the vestibular [80,81], visual [82–84], and somatosensory systems [85]. This duality reflects a trade-off between relatively linear integration and temporally precise detection of transients [86], which contribute complementary views of the stimuli.

For periodically modulated stimuli, tonic neurons are able to encode envelope shape at low rates with minimal distortion, while onset neurons entrain and report periodicity and flutter at moderate rates, as is seen in the inferior colliculus [76]. At very high modulation rates, the phase-locked system is overwhelmed and the stimulus may be better encoded by a transformation to rate in cortex [87,88]. For aperiodic stimuli, onset units report the timing of discrete events. The communication sounds of primates, bats, and other species are rich in temporal structure [89,90] typically at low to moderate temporal modulation rates [91]. In zebra finch caudomedial nidopallium (NCM), a higher-level auditory area important for recognition of songs, phasic neurons responded preferentially to rapid temporal features and were coherent with frequencies up to 20 to 30 Hz, whereas tonic neurons followed low frequency modulations [92]. In human speech, the phonemic temporal modulation rate is about 15 to 30 Hz (coinciding with the peak of Bu1 synchronization approximately 16 to 32 Hz), while syllables occur at a slower rate of 2 to 5 Hz [93]. Therefore, multiple modulation rate regimes may be best encoded by different neuronal types.

While RS units had the highest selectivity to vocalizations in terms of overall firing rate, the timing of spikes in Bu1 units could best distinguish between stimuli. As is the case for phase-locking in the auditory nerve [73], Bu1 units could be instantaneously responsive without an overall change in firing rate. They transformed dynamic vocalization stimuli into temporally sparse and precise sequences of spiking. Similarly, some sites in human auditory cortex detect acoustic edges, encode rate of change, and transform speech into a series of discrete landmark events [18]. In zebra finch NCM, excitatory neurons encode song sequences with temporally sparse and precise spike trains [30]. Temporal codes can convey information regarding vocalization identity [94], allow finer decoding of modulation rates [95], and be robust to background noise [30].

To test whether these codes can be read out on the population level, we implemented a population decoder that uses fine temporal information, and found that bursting unit populations could achieve high accuracy with a relatively small population size. This performance was more impaired by collapsing over time bins than by collapsing over units. However, both bursting and RS unit decoding performance decreased when temporal information or unit identity was lost, suggesting that both unit types contained temporal and labeled-line information. The optimal decoding time bin was smaller for bursting units than RS units.

These results concur with a previous study of macaque auditory cortex with natural sounds where responses were also considered in fine time bins [31]. It was found that while population decoding performance generally improved with the size of the population, a small ensemble of highly informative units could convey as much information as a large ensemble of randomly sampled units. Such highly informative units could be identified by the high temporal

precision of their responses, and could correspond to the bursting units we observed. Therefore, when interpreting neural activity in research or neural prostheses, it is important to analyze the results with sufficient temporal resolution. Furthermore, we should advance beyond the notion that units behave and contribute identically in population decoding. For instance, preferentially decoding temporally precise units or using different time bin sizes for the different unit types may more result in more accurate and efficient decoding. The apparent superiority of bursting units for decoding may reflect the situation of sparsely sampling from the full population of neurons. If RS units are highly selective in their overall rate, they may be harder to drive with any given stimulus, and conversely the most strongly driven neurons for that stimulus may be missed by the recordings. Therefore, we do not conclude that bursting units are necessarily more informative, but they are distinctly informative.

In our experiments, the animal was not required to perform any perceptual task. Previous studies have shown that perceptual task demands can lead to plasticity of neural population properties or even rapid plasticity of receptive field of individual neurons in auditory cortex [96]. In the context of a temporal task or when rapid pulse trains are paired with basal forebrain stimulation, plasticity can be seen in temporal response properties [97–99]. There are multiple ways in which unit type might interact with plasticity effects—particular types may be more prone to temporal plasticity, or alternatively but not exclusively, excitability of the various types may be reweighted in favor of temporally privileged types. Future studies should evaluate the relationship between unit type and behavioral task or social context in auditory and vocalization encoding.

Temporally precise onset type responses may additionally create temporal reference frames to align responses [44], entrain oscillatory processes to ongoing speech [100,101], contribute features for speech intelligibility in noise [102,103], segregate streams based on temporal coherence [104,105], and mediate gap detection [106]. They may be impaired in auditory processing disorders and dyslexia [107–109], autism [110], or aging [111]. Lastly, given these differences in coding between unit types, neuroprosthetic interfaces in the auditory system [112] and beyond [5,113] may benefit from considering unit type if single-neuron resolution can be achieved.

## Materials and methods

### Experimental model and subject details

Two adult male marmoset monkeys (ages 4 and 5, both around 380 g) were used in this study. Experimental animals were housed individually within spacious cages in a colony with audio-visual access to other conspecifics. Each animal was provided with enrichment toys, foraging mats, and a nest box. Animals were given water ad libitum, fed with LabDiet formulated for New World primates, and supplemented with food enrichment multiple times a week. Animals were gradually acclimated to sitting in a custom marmoset chair in a soundproof chamber. Once properly adapted, they were surgically implanted under general anesthesia with a headpost and chronic recording chambers, as described in previous publications from the lab [32]. Animals were monitored continuously during and after the procedure and given buprenorphine to alleviate pain during recovery on a temperature-controlled hot water pad under video observation. Recordings were collected chronically, while the marmosets listened passively to presented sounds. When necessary, animals were killed using a medical grade pentobarbital-based euthanasia solution (Euthasol, Virbac, Westlake, TX), and perfusion was initiated after cessation of the heartbeat. All procedures were approved by the Johns Hopkins University Animal Use and Care Committee.

### Method details

**Electrophysiological recordings.** Single-unit tungsten recordings were obtained from the left hemisphere of both animals. Recordings were taken along the cortical surface adjacent to the lateral sulcus, comprising mostly core auditory cortex regions A1 (primary auditory cortex), R (rostral core), and RT (rostrotemporal core) with possible coverage of CM/CL (caudomedial and caudolateral belt) and anterior secondary regions (see S2 Fig). Tonotopic gradients (S2D and S2E Fig) were compared with the average gradient schematic shown in S2A Fig and with previous studies such as [41]. Signals were amplified by an AC amplifier (Model 1800; A-M Systems, Sequim, WA) and filtered at between 1 Hz and 5 kHz. Units were detected based on spontaneous firing during slow advancing of the electrode; no search stimulus was used. Action potentials were then triggered using a template-based spike sorter (MSD; Alpha Omega Engineering), Nof HaGalil, Israel, while a simultaneous raw signal was digitized at 24.414 kHz and also stored. A standard 5/octave tone tuning protocol was recorded at 48 dB SPL. Tuning was then refined up to 40/octave resolution (if needed). Intensity (in 10 dB increments up to 68 dB SPL) and bandwidth tuning (0.05 to 3.2 octaves logarithmically spaced) were then recorded. SAM and duration stimuli were delivered at best frequency and bandwidth and at the best sound level for nonmonotonic units or 30 dB above threshold for monotonic units. Vocalization stimuli were previously recorded from multiple marmosets [114] and assembled into a list of mixed vocalization types ("Mixed Vocalizations List") as well as lists of exemplars of the same type of vocalization ("Call Type Lists").

**Quantification and statistical analysis.** Data from both animals were combined, because qualitatively similar results were seen for each animals analyzed separately. Analysis was performed in MATLAB (MathWorks, Natick, MA) and Python. Analysis scripts can be viewed at gitlab.com/a5640/AC_neuron_temporal. Peak picking was based on the function *peakfinder* [115]. Violin plots were generated with the function *violinplot* [116]. Typically, the response window was from 10 ms after the onset of the stimulus until 50 ms after the offset of the stimulus. For the longer vocalization stimuli, the window was extended until 150 ms after the offset of the stimulus. PSTHs were smoothed by convolution with a Gaussian ($\sigma$ = 5 ms). Group differences were compared using Welch's ANOVA followed by the Games–Howell post hoc test because variances were typically not equal between groups, even after log transformation (Levene's test).

**Cell type features.** The first method of determining cell types involved the consensus of criteria on a set of features. The second method fit clusters after dimensionality reduction on a set of informative features (described in the next section). The features and their calculations are described here.

Histograms of the ISI or log(ISI) were created with a bin size of 0.2 ms or 0.1 in log units. *Refractory period* was calculated from the ISI histogram as the smallest ISI bin for which the spike count exceeded 1/200 of the peak height of the histogram. Note that for nonbursting units with low firing rates, the histogram was often sparse and noisy at short intervals and a suspiciously large value of refractory period can result. A true refractory period calculation may require a longer period of spiking data in those cases.

The *ISI peak* was calculated from the log(ISI) histogram in a truncated manner by only considering the histogram below 80 ms in order to avoid detection of a second peak at high ISI values created by taking the log, as well as by the stimulus repetition period of approximately 770 ms. ISI histograms under a Poisson assumption have an exponential decay that combined with refractory properties create a right-skewed peak over the shorter intervals. This peak can be hard to discriminate from the peak caused by bursting, especially for FS units with high overall rate. The log(ISI) histogram more easily distinguishes these cases, as noted in Nowak and colleagues [33], because taking the log of the ISI values creates a more symmetric default peak

shifted out according to the firing rate. For a Poisson neuron with firing rate approximately 50 Hz (ignoring refractory period), the log(ISI) has a peak located around 3. For one with firing rate approximately 1 Hz, the peak is around 7. For well-driven units, the firing rate is not stationary, and in certain cases the trial repetition rate may also appear as a peak approximately 6.5. Although many factors affect the log(ISI) histogram, its peak value provided a good feature for distinguishing bursting units, which had an additional peak at <2. This bimodality could be demonstrated by *Hartigans' dip test p-value* with sufficient intervals. The test was performed in MATLAB using the function HartigansDipSignifTest [117].

The *logISIdrop* was constructed to detect a short interval peak and sharp drop-off from the log(ISI) histogram below 16 ms. Since this is much shorter than the prestimulus period of 200 ms, we did not correct for the effect of segmentation. First, the whole histogram was smoothed with a polynomial method (Savitzky–Golay with a span of 5 and degree 3). Next, the maximum value was determined in the region between 1 and 5 ms. This maximum was compared with the mean value between 10 and 16 ms (5 points) as follows:

$$logISIdrop = \frac{\max\left(y_{(\log(1),\log(5))}\right) - \bar{y}_{(\log(10),\log(16))}}{\max\left(y_{(\log(1),\log(5))}\right) + \bar{y}_{(\log(10),\log(16))}}$$

where $y$ is the log(ISI) histogram. The measure normally ranges between –1 and 1, approaching 1 as the relative size of the peak increases. However, it can rarely go beyond these bounds if the smoothing fit assigns negative ISI values to some points. A minimum of 50 ISI intervals per unit was required to reduce inclusion of very noisy histograms with insufficient spikes. This would be expected to bias against units with very low firing rates. Initially, the entire recording was considered, including stimulated and unstimulated periods, and only the standardized tuning protocol was used. In a second iteration, bursting units were identified based on logISIdrop of only spikes that occurred during the prestimulus period ("PBu") in order to eliminate the influence of driven bursts. Since this unstimulated state was nominally similar across protocols, all available files for each unit were pooled, allowing very low spike rate units to potentially surpass the minimum number of intervals via pooling. If a unit was not labeled "Bu" but was labeled "PBu," the unit was considered "Bursting ambiguous" rather than "Non-bursting" and was not considered for further classification as "RS" or "FS."

The autocorrelogram metric compared the relative size of the mean of the autocorrelogram below 8 ms versus the mean of the autocorrelogram between 35 and 80 ms. These ranges avoid the potential gamma frequency peak reported for chattering neurons in the literature.

$$ACM = \frac{\bar{R}_{(0,8)} - \bar{R}_{(35,80)}}{\bar{R}_{(0,8)} + \bar{R}_{(35,80)}}$$

Intraburst frequency was calculated as the inverse of the non-log ISI peak at 0.2 ms resolution.

For spike waveform analysis, only spikes that occurred in isolation (no other spikes within the interval from 5 ms before to 6 ms after the current trigger) were included in the spike averages for cell type classification. A minimum of 5 spikes was required in this average. Spike waveform was less likely to be protocol dependent than spike timing, so we pooled up to the first 5 recordings from each unit. To minimize distortion, waveforms were filtered broadly at 1 through 10,000 Hz with a 256th-order FIR filter processed in forward and reverse to produce zero-phase filtering. Although this is wider than our amplifier settings, it provided a bit of additional smoothing. Spike waveforms were then aligned by the largest voltage change (up or down sweep, whichever was larger). The trough-to-peak time was calculated by first locating positive and negative peaks around the maximum slope point. The trough-to-peak time was

taken as the time between the largest downward peak and the next upward peak. Half-amplitude duration was calculated at halfway from the trough back toward baseline, with baseline estimated by time averaging a 1-ms interval immediately before the spike. The spike waveform was interpolated with a cubic spline to 10× resolution before calculating the half-amplitude duration.

We also performed frequency analysis on the spike waveform. The spectrum of the baseline period preceding the spike was subtracted from the spectrum spanning the spike. For this analysis, spikes were considered isolated if they were no other spikes in the interval from 10 ms before to 6 ms after the spike to provide a sufficiently long segment for analysis of the spike and baseline. The $f_{50}$ was defined as the high-side frequency at which the spectrum rolled off to 50% of its peak. Peak was determined above 400 Hz to avoid noisiness as the cycle length approaches the segment length. Although multiple full spikes were disallowed, it was not always possible to exclude mini spikes that followed in bursts and the spectrum could still reflect this as periodic peaks superimposed on the spike spectrum. The variability in how the 2 aligned may have contributed additional noise to the $f_{50}$ of bursting units.

For mean and max burst length, a burst was considered a group of consecutive ISI's that fell between 0.5 and 1.5 times the ISI peak, with a minimum length of 1 ISI, corresponding to 2 spikes. This has a natural interpretation for bursting units as the ISI peak corresponded to the intraburst interval. Although the interpretation of this definition is less obvious for FS and RS units, the high burst lengths for FS units agrees with the presence of relatively regular longer strings of spikes in a subset of units. This feature was inspired by the maximum number of spikes in a burst (NSB) in Katai and colleagues [27], which also found long bursts for FS-like units and short bursts for chattering-like units.

Previous studies have used the fraction or percent of ISI less than 5 ms to detect burstiness [27,39]. We also computed the percent normalized by that expected for a Poisson process with the same mean rate to account for differences due to firing rate [7,23,39]. Although these measures were much higher for bursting units (Fig 3), we did not use it as 1 of our 3 criteria for detection of bursting. However, it was included in the GMM classification. For the normalized measure, FS units had a mean of 1.29, suggesting they are close to a Poisson process, whereas bursting units had a mean of 46.7. A few outliers among RS units entered the range of the bursting units, possibly due to noisiness from having few spikes and intervals.

We defined a maximum "burst" length based on consecutive ISI's close to the peak ISI. Bursting units tended to fire only a few spikes per burst, whereas some FS units exhibited longer spontaneous trains of spikes, consistent with the observations of Katai and colleagues [27]. We considered using the coefficient of variation (CV) but felt that it would be artificially inflated by the inclusion of driven and nondriven periods [118]. Since CV is likely not the most sensitive way to detect bursting, we did not pursue it further. However, an adapted approach is to examine only adjacent ISIs that reduces the effect of changing rate [119]. Based on Parikh and colleagues [120], we calculated a similar notion, termed the regularity (or rather irregularity), as the variance of a ratio of consecutive ISI values ($\frac{ISI_n}{(ISI_n+ISI_{n+1})}$). This measure is indeed highest for bursting units and is lowest for FS units (S3 Fig).

Also inspired by Parikh and colleagues [120], we tested a burstiness measure defined as the fraction of ISI less than 5% of the mean ISI. This was not as different for the various groups as we had hoped (S3 Fig), perhaps because ISI distributions are generally right skewed.

**Unsupervised classification cluster analysis.**   We selected 8 features, namely the autocorrelation metric, ISI peak, logISIdrop, percent of ISI less than 5 ms, spontaneous rate, $f_{50}$, maximum burst length, and maximum firing rate across stimuli (averaged over the whole response window). Other similar sets of features might work as well and shorter feature lists seemed to

work but with less stability across random initializations. Features were log transformed if they were strongly skewed (often seen with features that cannot go below 0)—a tiny offset less than the minimum non-zero value was added if needed to avoid taking the log of 0. Each feature was standardized by removing the mean and dividing by the standard deviation to prevent large valued parameters from dominating. The data was then processed through a PCA, and a GMM with "full" covariance was fit to the first 3 PCA components. The PCA used the alternating least squares algorithm for missing data, but similar results were obtained using the "pairwise" method, and only complete rows were projected and used for the GMM. The optimal number of clusters was suggested by considering values of the AIC and BIC, and by mean negative log-likelihood (Fig 3D and 3E). To calculate likelihood cross-validation, half the data was used as a "training" set and half as a "test" set, and the procedure was repeated 50 times each for 1 through 7 components. Convergence depended on the choice of random seed, so the GMM was run for 20 consecutive random seeds, and the 1 with the lowest corresponding AIC was chosen. As units in the overlapping region may not clearly belong to 1 group versus another, we used the conservative criteria that only units whose posterior probability of being in 1 group was at least twice that of either of the other groups was assigned to that group.

**Analysis of duration responses and sinusoidally amplitude modulated (SAM) stimuli.** To quantify adaptation, we created an adaptation index that compared the rate during the first 100 ms of the response ("early") with the rate during the last 100 ms of the response ("late") (offset by 10 ms from the start of the stimulus):

$$AI = \frac{r_{early} - r_{late}}{r_{early} + r_{late}}$$

Early and late have slightly varying definitions in the literature—ours is similar to the definition of Recanzone and colleagues [121]. For analyses of PSTH, responses were tallied in 1 ms bins and convolved with a Gaussian with $\sigma = 5$ ms.

For a subset of units, we presented 100% modulation depth SAM stimuli at 2 through 512 Hz with the carrier determined by the best frequency, level, and bandwidth. The VS was calculated excluding the first 50 ms of response, and therefore would not consider a purely onset response as being synchronized. Nonsignificant VS were set to 0 to suppress noise, but this also somewhat distorts the averages. However, excluding nonsignificant VS values would artificially elevate the population VS. Nonsignificance could be due to poor phase-locking or insufficient number of spikes, but RS and bursting units had similar rates of spiking so this should not account for the much higher incidence of nonsignificant VS in the RS population. Rate response was calculated based on our standard window of 10 ms after stimulus onset to 50 ms after stimulus offset and required exceeding 3 SDs above the spontaneous rate. The significance of the VS was assessed by the Rayleigh statistic $2 VS^2 N$, where N is the number of spikes [122]. Rayleigh values greater than 13.8 were considered as significant, corresponding to $P < 0.001$ [11], and those that were not significant were set to 0. The maximum synchronization frequency ($f_{max}$) was determined by linear interpolation between the highest SAM rate with Rayleigh value greater than 13.8 and the frequency above that one. The frequency where the interpolation crossed 13.8 was considered the $f_{max}$. For units with at least 2 significant values of Rayleigh statistic, tBMF was calculated as a weighted geometric mean of the maximum VS and up to 1 adjacent value on each side, if significant. Following the convention of Bendor and Wang [41], a unit was deemed synchronized if it had a significant Rayleigh statistic and VS >0.1 for at least 1 modulation frequency between 4 and 512 Hz. A second calculation was also performed with the more stringent requirement that this be true for any modulation rate between 16 and 512 Hz. Units without significant Rayleigh values, but that did have significant

rate responses in the range of modulation rates considered, were deemed as unsynchronized units.

**Correlation index (CI).**　Vocalizations were presented randomly interleaved for 10 repetitions each. Shuffled autocorrelograms were calculated as in Joris and colleagues [123]. Briefly, for each neuron and stimulus, all-order ISI histograms were constructed between all pairs of spike trains except trains with themselves (S8A and S8B Fig). This procedure detects the tendency for spikes to occur at the same time but bypasses the confounding effect of the refractory period. The CI is normalized to account for the predicted effect of firing rate, stimulus duration, number of repetitions, and choice of coincidence window, facilitating comparison across unit classes. Based on the falloff of CI values with increasing temporal window size (10 log-spaced samples per order of magnitude), we computed the CI based on the average of the 5 window sizes flanking 0.5 ms (S8C Fig). To assign a single CI value per neuron, we used the maximum CI across all responsive stimuli ($CI_{max}$), where responsive was taken to mean significantly excited either by average rate over the entire stimulus response window or by maximum PSTH values (5 ms bins). To determine if a unit was responsive to a particular stimulus, we looked at the distribution of rates during the prestimulus period (across all stimuli and repetitions). Under the null hypothesis, the mean rate during the stimulus (averaged across repetitions) should fall within this distribution; a response was considered significant if it exceeded the mean plus 3 standard errors (for the number of averaged repetitions). The variance of the rate should approximately scale inversely with the duration, so a correction was applied as the vocalization stimuli can be much longer than the prestimulus period. Note that only excitatory responses are considered as it may not make sense to analyze spike timing in strongly inhibited responses. To calculate whether PSTH values were significantly responsive, the number of spikes in each 5 ms time bin was aggregated across repetitions. This distribution would be expected to be roughly Poisson and strongly skewed, so we fit the histogram of counts from the spontaneous period with a Poisson distribution which was then used to calculate probabilities. PSTH's were considered responsive if the maximum PSTH value during the response period corresponded to a $p$-value of $<0.01$ with Bonferroni correction for the number of response PSTH bins.

**Spike train decoding.**　Several metrics have been proposed for studying the importance of spike timing in stimulus decoding [124]. For pairs of spike trains, the Victor–Purpura [43] and van Rossum metrics [125] assign a spike distance that has formal properties suitable for a Euclidean distance metric, can be computed efficiently, and span from a rate code to increasing precision in spike timing information. These computed distances can then be used to classify spike train responses to various stimuli. As a single time scale parameter ($q$) is varied, the effect on the quality of classification can be assessed by the transmitted information $H$ of the confusion matrix. The Victor–Purpura distance calculates the total "cost" of transforming 1 spike train into another. Adding or deleting spikes each is associated with a cost of 1, while shifting spikes by $\Delta t$ has a cost of $q|\Delta t|$. Where $\Delta t$ exceeds $2/q$, it becomes more cost effective to delete and reinsert the spike than to shift it. For $q = 0$, spikes can be shifted at no cost so the metric is independent of spike timing and equal to the difference in the number of spikes. For large values of $q$, spikes in 1 train can only be cost-effectively shifted by a small interval in order to match the other train and the code demands high temporal precision. Spike distances were computed using code in MATLAB [126]. For the classification of each spike train, the train itself is excluded from the spike train pool, and distances to all other spike trains are computed. The spike train is then assigned to the stimulus that has the minimal average distance across repetitions. Per the original method [43], we used a power transformation with an exponent of $z = -3$ in the averaging step that emphasizes small distances. In the case of tied minimal distances, the assignment of that spike train was tallied as $1/k$ for each of the $k$ tied stimuli. The

distribution of chance performance was computed by shuffling the stimulus labels of the spike trains 100 times, and units whose classification performance exceeded a $z$-score of 3 were pooled for this analysis. Since rate coding can distinguish between stimuli that produced a response and stimuli that did not, we performed classification on the full set of stimuli rather than only the subset of responsive stimuli as we did for CI.

An issue in applying these metrics to our data is that vocalizations vary considerably in length that can produce artifactual effects. For instance, it is possible to classify spontaneous spike trains of different lengths simply based on the total number of spikes. When spike timing is taken into account, spikes beyond the duration of the shorter stimulus would inevitably incur a large temporal cost. Therefore, for each vocalization stimulus, we chose a response segment equal in length to the shortest stimulus and centered on the peak of the kernel density estimate of the response. This recentering removes some timing information, especially if the response has only 1 cluster of spikes, rather than 2 or more, and reduces differences in rate between stimuli. We also observed as others have [127] that the power transformation produces some distortions and particularly impairs rate coding; the average distance of any set that contains a distance of 0 is mapped to 0 by this transformation, and this occurs when $q = 0$, and the spike counts are the same or when repetitions with no spikes are compared at other $q$ values. Lastly, these metrics contain implicit count and rate information and are not only sensitive to spike timing [128]. In comparison, the Schreiber distance [129] measures reliability, but is sensitive to missing or additional spikes as well as temporal precision, and is not ideal for spanning to rate coding because it is a correlational metric that normalizes for rate. Despite the imperfections mentioned above, the Victor–Purpura distance metric does show the relative performance of decoding as the temporal precision requirement is increased from a rate code to a very temporally precise code. Note that rate and temporal information are not mutually exclusive—a temporally precise neuron could provide plenty of information for rate decoding if it responds to only a few stimuli or if, for example, the neuron fires in a precise manner but only for a particular direction of FM sweep.

**Population decoding.** A simple classifier based on the maximum correlation of test responses with the mean stimulus responses can perform similarly to more complicated methods such as support vector machines or naïve Bayes decoding [130]. This method creates a mean response vector for each stimulus class then maps test trials to the class with which it has the highest correlation coefficient [131]. For our categorical decoding of time-binned population responses, we had success with the maximum correlation decoder as well as with LDA without a prior and with shrinkage regularization. The latter achieved similar performance but required much longer processing times likely due to pairwise calculation of covariance matrices for the large number of features. The response of each unit in each trial was quantified in M nonoverlapping 10 ms time bins for a population of N units. Vocalization stimuli were of varying length, so responses were cropped to the duration of the shortest stimulus plus 300 ms to create equal-length response vectors (responses for short stimuli included some poststimulus time, similar to zero-padding). As units were not recorded simultaneously for the most part (although occasionally up to 3 units were isolated at the same site by spike sorting), we used a randomization process to create pseudopopulation responses. Each pseudopopulation response to a particular stimulus consisted of M × N features where each feature was the response of a particular unit in a particular time bin. Percent accuracy was calculated as the sum of the diagonal values divided by the sum of all values in the confusion matrix. More units were available for this decoding than for $CI_{max}$ because some units did not have any intervals within the coincidence window used to calculate CI. For population decoding, units were only required to have at least 1 responsive stimulus, as assessed by rate or PSTH, whereas for the Victor–Purpura metric-based classification, each unit had to individually achieve a statistically

significant decoding performance to be included. Unit types were determined by the method of criteria. Decoding was implemented in Python.

## Supporting information

**S1 Fig. Bursting could be seen in the full response as well as during the unstimulated period.** (A) Raster plot of an example bursting unit (M117B0636ch4) in response to tones. A short-latency transient response is seen at 5.3 kHz. Rapid and brief bursts occurred throughout the prestimulus, stimulus, and poststimulus periods, with 2 expanded bursts shown in the red and blue boxes. (B) Autocorrelogram (0.2 ms resolution) calculated from the entire response shows bursting behavior with a peak at 1.1 ms. (C) Autocorrelogram calculated from only the prestimulus periods is also maximal at 1.1 ms. (D) A 100 second-long segment of spontaneous activity was also recorded for this unit and the same spike timing properties can be observed, with a peak at 1.1 ms. Data underlying this figure can be found in S2 Data.
(TIF)

**S2 Fig. Unit types did not differ grossly in terms of frequency, depth, or regional distribution despite showing consistent functional differences.** (A) Schematic showing location of auditory cortex along the lateral sulcus of the left hemisphere of the marmoset brain and cortical areas within the core region (dark shading) and belt region (light shading), based on (1–3). Within the core, the tonotopy experiences a low frequency reversal at the lateral border between AI and R, and a mid-frequency reversal at the medial border between R and RT. Recordings were made along the length of the lateral sulcus, primarily in core areas AI, R, and RT, with some likely inclusion of anterior and caudal belt. (B and C) BF and recording depth distributions were generally overlapping for the various unit types (RS, *red circles*; FS, *blue squares*; Bu1, *dark green triangles*; and Bu2 *light green diamonds*), and should not be a confounding cause for consistent unit type differences observed between unit types. Depths are expressed relative to the first spiking unit encountered from a superficial approach and were biased toward superficial layers due to the long recording times spent with each unit. One-way ANOVAs did not show a statistically significant difference in BF or depth between at least 2 groups ($F(3,329) = 1.97$, $p = 0.12$ and $F(3,355) = 1.6$, $p = 0.19$). (D andE) Maps of best frequencies of recorded units in the 2 marmosets used in this study, spanning from the low frequency region of anterior RT to the high frequency region of posterior AI. See (H and I) for scale. A small jitter was added to offset multiple units within the same track for visibility. Light gray x's indicate units that could not be well driven by sound. (F and G) Unit types were distributed throughout recorded areas. For instance, Bu1 units (dark green triangles) were interleaved with other unit types. (H and I) When units were projected onto the sulcal axis, bursting units, and in particular Bu1 units, had higher CImax values regardless of anterior-posterior location. Data underlying this figure can be found in S2 Data. AI, primary auditory cortex; AL, anterolateral belt; BF, best frequency; CL, caudolateral belt; CM, caudomedial belt; FS, fast-spiking; ML, middle lateral belt; MM, middle medial belt; R, rostral core; RM, rostromedial belt; RS, regular-spiking; RT, rostrotemporal core; RTL, rostrotemporal-lateral belt; RTM, rostrotemporal-medial belt;
(TIF)

**S3 Fig. Unit types differed in terms of a number of properties.** The first 6 properties were used for classification by criteria and the criteria boundaries are shown in gray lines (see Methods). The other properties were not used in making the classification and include basic properties and additional properties we explored for identifying bursting (see Methods). Unit type was determined by criteria, or solely based on prestimulus logISIdrop for PBu. Bu1 and

Bu2 are also shown separately. The consensus criteria meant that the cutoff for a single property was often "soft," as evidenced by the tails of some distributions crossing over the dividing lines. FS units had (1) spikes with shorter $t_{\mathrm{TTP}}$ and larger $f_{50}$ values; (2) higher spontaneous and maximum driven rates to tones; (3) clearly unimodal ISI histograms according to Hartigans' dip test; and (4) a propensity for firing strings of spikes (high max "burst" length, with burst defined as consecutive ISI values between 0.5 and 1.5 times the mode of the ISI). Bursting units were characterized by (1) very short peak ISI values reflecting the bursting interval; (2) differences in the autocorrelogram metric, logISIdrop, and the percent of ISI less than 5 ms; (3) indication of bimodality on Hartigans' dip test; and (4) bursts with a smaller max and mean burst length (fewer spikes per burst). RS units had (1) long $t_{\mathrm{TTP}}$ and lower $f_{50}$ values; (2) relatively long calculated refractory periods; and (3) longer minimum response latencies. Compared with Bu2 units, Bu1 units had higher values of the autocorrelation metric and logISIdrop, narrower spikes, shorter refractory periods, and shorter latencies. Three outliers with very large refractory periods are cropped out. Maximum firing rate was the maximum mean rate during a stimulus response window. Data underlying this figure can be found in S2 Data. FS, fast-spiking; ISI, interspike interval; RS, regular-spiking.
(TIF)

**S4 Fig. Properties of Bu1 and Bu2 subgroups as separated by GMM clustering are similar to those of bursting subgroups as determined by the 500 Hz intraburst frequency criteria.** (A) Responses to 400 ms long synthetic stimuli in Bu1 units have shorter latency, higher peak firing rate, and more complete and rapid adaptation than responses in Bu2 units. (B) Bu1 unit maximum firing rate was sensitive to the rate of sound onset. (C) Bu1 unit VS was higher than Bu2 VS and peaked at intermediate SAM rates. (D) A majority of Bu1 units were synchronized at 16 Hz or higher SAM rate, in contrast with RS, FS, and Bu2 groups. (E) CI was highest for Bu1 units, indicating a tendency for spikes to occur at nearly the same time on each repetition of the vocalization stimuli. (F) This tendency is also reflected in the Bu1 group's right shifted (toward temporal encoding) $H$ versus $q$ curve for decoding based on the Victor–Purpura spike distance metric. Data underlying this figure can be found in S2 Data. CI, correlation index; FS, fast-spiking; ISI, interspike interval; PSTH, peristimulus time histogram; RS, regular-spiking; SAM, sinusoidal amplitude modulation; VS, vector strength.
(TIF)

**S5 Fig. Response properties to SAM, classified by GMM and all bursting units.** Same plots as Fig 6, but using unit type labels from the clustering analysis rather than the labels generated by criteria. Bursting units identified by 3 methods are shown for comparison: Bu (from GMM), PBu (from prestimulus logISIdrop), and Bu$_{\mathrm{crit}}$ (from method of criteria, Bu1 and Bu2 combined). (A) Violin plot of maximum VS for RS (74), FS (65), Bu (40), PBu (45), and Bu$_{\mathrm{crit}}$ (35) units. (B) Mean VS versus SAM modulation rate. (C) Violin plot of maximum synchronized rate for each unit type. (D) Fraction of responsive units that were synchronized at or above 4 Hz. (E) Fraction of responsive units that were synchronized at or above 16 Hz. (F) Average period histograms for stimulation at 2 Hz. Data underlying this figure can be found in S2 Data. Bu, bursting; FS, fast-spiking; GMM, Gaussian mixture model; RS, regular-spiking; SAM, sinusoidal amplitude modulation; VS, vector strength.
(TIF)

**S6 Fig. Spectrograms of "Mixed Vocalizations List" and "Call Type List" stimulus panels.** (A) The standard "Mixed Vocalizations List" included 10 call tokens in natural ("nat") or and time-reversed ("rev") orientation. For detailed descriptions of call types and compound calls, refer to [114]. In some cases, we also played lists of example tokens of the same vocalization

type, such as the "Phee Call Type List" (B) and "Trill Call Type List" (C).
(PNG)

**S7 Fig. Responses of bursting units to the "Mixed Vocalizations List" (S6A Fig).** Examples of diverse precise responses to vocalizations from bursting units (intraburst frequency shown in top right corner). Alternating light aqua shading indicates the stimulus duration. Data underlying this figure can be found in S2 Data.
(TIF)

**S8 Fig. Calculation of CI.** (A) The SAC was calculated as the all-order ISI histogram between spikes in 1 repetition and spikes in all other repetitions of that stimulus. "Shuffling" by excluding within-trial intervals removes the effect of the refractory period and direct effects of bursting. (B) An example of the SAC calculated from the response of a bursting unit to a marmoset *trill* vocalization, cropped to show short time scale autocorrelation. There was a strong tendency for spikes to occur within milliseconds of each other in the stimulus time frame across repetitions. (C) From the SAC, we can calculate the CI, a normalized measure of the prevalence of "coincidences," or intervals smaller than a particular coincidence window ($\omega$) [42]. For very small coincidence windows, we see a higher level of noise. For large windows, the coincidence "density" falls off. We chose to calculate the CI as the average of the 5 values around $\omega = 0.5$ ms (black arrowhead). The CI measures the tendency for spikes to occur at the same time(s) within the stimulus, can be seen as a generalization of VS to aperiodic stimuli, and is scaled to account for firing rate, stimulus duration, number of repetitions, and $\omega$. Data underlying this figure can be found in S2 Data. CI, correlation index; ISI, interspike interval; SAC, shuffled autocorrelogram; VS, vector strength.
(TIF)

**S1 Data.** "S1_Data.xlsx" includes the data underlying the main figures.
(XLSX)

**S2 Data.** "S2_Data.xlsx" includes the data underlying the Supporting information figures.
(XLSX)

## Acknowledgments

We thank Jessica Lynch, Sami Miller, Zach Schmidt, Kayla Schonvisky, Jessica Izzi, and the veterinary staff for technical and animal care assistance and Gregory Hale for comments on the manuscript.

## Author Contributions

**Conceptualization:** Xiao-Ping Liu.

**Formal analysis:** Xiao-Ping Liu.

**Funding acquisition:** Xiaoqin Wang.

**Investigation:** Xiao-Ping Liu.

**Supervision:** Xiaoqin Wang.

**Validation:** Xiaoqin Wang.

**Visualization:** Xiao-Ping Liu.

**Writing – original draft:** Xiao-Ping Liu.

**Writing – review & editing:** Xiao-Ping Liu, Xiaoqin Wang.

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
