## [Editor Report · Decision Letter 0]

29 Nov 2021

Dear Dr Liu, 

Thank you for submitting your manuscript entitled "Neuronal types contribute to hybrid temporal encoding strategies in primate auditory cortex" for consideration as a Research Article by PLOS Biology.

Your manuscript has now been evaluated by the PLOS Biology editorial staff, as well as by an academic editor with relevant expertise, and I am writing to let you know that we would like to send your submission out for external peer review.

Once your full submission is complete, your paper will undergo a series of checks in preparation for peer review. Once your manuscript has passed the checks it will be sent out for review. To provide the metadata for your submission, please Login to Editorial Manager (https://www.editorialmanager.com/pbiology) within two working days, i.e. by Dec 01 2021 11:59PM.

If your manuscript has been previously reviewed at another journal, PLOS Biology is willing to work with those reviews in order to avoid re-starting the process. Submission of the previous reviews is entirely optional and our ability to use them effectively will depend on the willingness of the previous journal to confirm the content of the reports and share the reviewer identities. Please note that we reserve the right to invite additional reviewers if we consider that additional/independent reviewers are needed, although we aim to avoid this as far as possible. In our experience, working with previous reviews does save time. 

If you would like to send previous reviewer reports to us, please email me at ggasque@plos.org to let me know, including the name of the previous journal and the manuscript ID the study was given, as well as attaching a point-by-point response to reviewers that details how you have or plan to address the reviewers' concerns. 

Given the disruptions resulting from the ongoing COVID-19 pandemic, please expect some delays in the editorial process. We apologise in advance for any inconvenience caused and will do our best to minimize impact as far as possible.

Kind regards,

Gabriel

Gabriel Gasque

Senior Editor

PLOS Biology

ggasque@plos.org

---

## [Decision Letter · Decision Letter 1]

11 Jan 2022

Dear Dr Liu,

Thank you for submitting your manuscript "Neuronal types contribute to hybrid temporal encoding strategies in primate auditory cortex" for consideration as a Research Article at PLOS Biology. Your manuscript has been evaluated by the PLOS Biology editors, by an Academic Editor with relevant expertise, and by three independent reviewers.

In light of the reviews (below), we will not be able to accept the current version of the manuscript, but we would welcome re-submission of a much-revised version that takes into account the reviewers' comments. We cannot make any decision about publication until we have seen the revised manuscript and your response to the reviewers' comments. Your revised manuscript is also likely to be sent for further evaluation by the reviewers.

We expect to receive your revised manuscript within 3 months.

**IMPORTANT - SUBMITTING YOUR REVISION**

Your revisions should address the specific points made by each reviewer with additional analyses where requested. We do not think your revision needs extra data collection. Please submit the following files along with your revised manuscript:

*Re-submission Checklist*

*Published Peer Review*

*PLOS Data Policy*

*Blot and Gel Data Policy*

Sincerely,

Gabriel Gasque

Senior Editor

PLOS Biology

ggasque@plos.org

REVIEWS:

Reviewer #1: Overview

This study characterizes response properties of single neurons in the auditory cortex of awake marmosets, and concludes that there are two distinct functional classes, Regular Spiking and Bursting cells, the latter displaying superior encoding of rapid temporal features. The strength of the study is the breadth of descriptive and quantitative physiological assessments. However, significant weaknesses are that neural responses to natural stimuli (i.e., vocalizations) are poorly described, the data is pooled across many functionally distinct regions of auditory cortex, and the absence of a population analysis that evaluates how each sub-population contributes to encoding. These shortcomings can be addressed without the need for additional data collection.

Main comments:

(1) The manuscript, as written, is narrowly directed to the field of auditory neurophysiology, and may not be of interest to a broader audience. Therefore, a much broader rationale for these studies should be added to the Introduction. If possible, the experiments should be motivated by a strong hypothesis. As written, the main objective is to characterize spike categories and response properties, which comes across as a somewhat incremental advance over the authors' previous publications on temporal versus rate coding in auditory cortex.

(2) Since different regions of the auditory cortex are known to have different response properties (e.g., Bendor & Wang, 2008), it is essential to divide the data by electrode recording location (lines 431-2: A1, R, RT, CM/CL, anterior secondary regions). Of equal importance is the recording depth and putative laminar distribution. It is possible that the distribution of response classes, especially the Bu cells, differ significantly between auditory cortical regions and/or depths which could alter the interpretation.

(3) A full and clear description is needed for the vocalization portion of the study:

(a) As presented, four different call types are shown (phee, trill, twitter, tsik) in Supplementary Figure 5, but only responses to "phee" and "trill" appear to be presented (Figure 6), or discussed in the text. Either clarify where each call type is presented, both in the text and in the Figure legends, or remove calls that were not used in this study.

(b) Which call types are shown in Figure 7 and Supplementary Figure 6? The vocalization used for every raster should be clear to the reader. It would be helpful to see the firing rate for each response type to the four call types (i.e., spectrogram of call with associated rasters).

(c) Figures 6, 7, and S6 are difficult to interpret, especially given the absence of a full description in either the text or the Figure legends. Are the light aqua/grey bands indicating call duration? For Figure 6, what do each of the light aqua/grey bands represent (does each band correspond to 10 reps of a different phee call)? Are neurons recorded in the same cortical region? The same putative layer?

(d) It is difficult to distinguish the alternating rasters for the normal and the reversed calls (Figure 7D, Figure S6).

(4) All of the recordings are collected passively. Given that marmosets are a highly social species, and that auditory cortical responses are known to be modulated by active engagement in an auditory behavioral task, it is not clear how to interpret the data, especially whether the distribution of cells types and responses would differ while animals were engaged in a social interaction, or were actively discriminating between stimuli. Although the data set will not change, it is important that the Discussion present an evaluation of this issue.

(5) To better assess the contribution of each response type, it would be valuable to implement a population decoder. This would permit the one to include or exclude RS or Bu cells, and test for call categorization. It is not clear from the current analysis how well each class, alone, would perform. Furthermore, given that FS cells are putative inhibitory interneurons, and would not be expected to contribute to downstream decoding, a population decoder would not include this response type. An analysis of this sort would help the manuscript shift from a descriptive study to a more hypothesis-driven one.

Minor comments:

(1) Write out abbreviations in Methods and Details section. For example: Line 431: A1, R, RT, CM/CL; Line 439: SAM; Line 573: AIC, BIC; Line 544: GMM

(2) Methods: Would be helpful to define best frequency and bandwidth tuning for the general reader

(3) Figure 1: Panel "D" is mislabeled as "B"

(4) Figure 1: The label for Panel "E" is missing

(5) Figure 1: Some of the panels are missing y-axis units. E.g., For panel A, what is the y-axis? Trial number?

(6) Figure 4: What are the units for the heat map? Firing rate?

(7) Supplementary Figure 1A: Missing x-axis units and labels

(8) For the raster plots, it's difficult to see the gray bars, especially when showing the responses to 'reversed' vs 'normal' vocalizations (e.g., Figure 7)

(9) Figure 7: Write out what the "r" and "n" is in the figure legend

(10) Supplementary Figure 5: Which spectrograms correspond to which call types? Labels would be helpful.

Reviewer #2: In their manuscript entitled "Neuronal types contribute to hybrid temporal encoding strategies in primate auditory cortex", Liu and Wang investigate whether different subsets of neurons - approximately classified by waveform/activity properties - in the auditory cortex have different roles in encoding sounds, in particular vocalizations. The manuscript is approaching an interesting scientific question using rigorous methodology and is executed very well, both on the level of the writing and illustrations. There, however, remained a number of important questions to be addressed (see below) which may require substantial additional analyses.

1) The Methods section omits information about the detailed recording procedures, pointing to previous published work. While this is generally fine, for the present results it would be highly relevant to know the location of recordings, in particular in which areas/subareas were the different neuronal types localized? Was there any area-related grouping of the units, or were they distributed homogeneously among the areas? Was there any relation to preferred frequency/tonotopy? Was there any preferred depth/layer where the different groups of neurons were localized? The revised version should include an analysis of the response properties/classifying metrics/classes as a function of area/layer/tonotopic location, to relate the current findings to previous work.

2) Relation to other response properties: Currently, the analysis/manuscript is focussed on the specific response properties of the units, however, does not relate these results to other, more basic properties, in particular classical characterizing metrics such as STRFs, which would link the current results better to previous work. Given the time-locked response properties of the Burst neurons, an average, CF centered STRF of the groups would likely show that many clear STRFs are originating from the bursting groups. This would help explain the varying levels of SNR typically found in STRFs from auditory cortex.

3) Scientists tend to classify their results in order to simplify the results. While this is a useful practice, it sometimes can mistake a continuum for a set of classes. While the presented data leave no doubt about a wide range of expression of certain response properties (i.e. bursting phenotype/phase-locking in this case), it remained in my opinion inconclusive whether there are really identifiable classes. As the authors note themselves, " Therefore, if bursting and non-bursting units were combined, the resulting histogram may not appear bimodal", and if I interpret this sentence and the graphs correctly, the neurons that remained unclassified would in addition fill the putative dip between the putative classes (see also Fig. 2B/3B). The logic for using the dip test was inverted here (applied after selecting classes), which - given the current criteria - would be biased to find dips in many single-class or continuous distributions. By eye, if the marginal histogram in Fig. 2B included the non-classified units, I would estimate it would not show a (significant) dip. I think the authors should therefore (1) either not make the claim that these units are 3 separate classes and instead argue for different response properties of units along a continuum, (2) provide stronger support for separable classes of neurons, e.g. either by genetic identification/biochemical labelling or identifying a combination of dimensions where the projected density of response properties passes the dip test (this might also be the case for the Fig. 3B, but I could not find a corresponding claim/analysis in the text).

In association with the question above, the authors could pursue the question of whether classes or a continuum would be better suited for en-/decoding of naturally occurring sounds/vocalizations (likely continuum). While this appears to be outside the scope of the present article, it would provide a strong motivation for the existence of the observed range of response/coding properties.

4) Availability of the Data/Analysis Code: In their submission information, the authors indicate that the data is fully available, however, no additional information is provided. In particular during the review process the availability of the (near raw) data and the analysis code would be useful. It could e.g. be made available to reviewers confidentially during the review process, and later openly to all readers.

5) Bursting or 'Doublet Neurons': It was hard for me to assess what the number of spikes in a burst typically was, from the information in the manuscript. Are these dominantly doublets or triplets or even longer sequences? To clarify this, the revision should include a histogram of the number of spikes in a burst, and then relate this finding to bursting cells in other systems (extending/modifying the excellent discussion on this topic).

Minor:

- Please include the Bu1/Bu2 grouping into Fig. 2.

- Please include the marginal across all units in Fig. 2B (right subpanel)

- An interesting general reference for the introduction: https://pubmed.ncbi.nlm.nih.gov/30034330/

Reviewer #3: No comments (accept).

---

## [Editor Report · Decision Letter 2]

11 Apr 2022

Dear Dr Liu,

Thank you for submitting your manuscript "Neuronal types contribute to hybrid temporal encoding strategies in primate auditory cortex" for consideration as a Research Article by PLOS Biology. Your revision was evaluated by the PLOS Biology editors as well as by an Academic Editor with relevant expertise. In this case, the Academic Editor felt comfortable evaluating your revision so it was not sent back out to the original reviewers. 

Based on our evaluation, we will probably accept this manuscript for publication, provided you satisfactorily address the remaining points raised by the reviewers. Please also make sure to address the important data and other policy-related requests at the bottom of this email.

1) Please change your title to read:

Distinct neuronal types contribute to hybrid temporal encoding strategies in primate auditory cortex

2) Please update your Ethics statement as per journal policy (details below).

3) Please provide all of the necessary raw data files as per journal policy, and update your manuscript figure legends to indicate where this raw data can be found. (Details below).

We expect to receive your revised manuscript within two weeks. 

*Published Peer Review History*

*Press*

Sincerely,

Kris

Kris Dickson,

Neurosciences Senior Editor/Section Manager,

kdickson@plos.org,

PLOS Biology

ETHICS STATEMENT:

PLOS Biology guidelines for Non-Human Primates:

https://journals.plos.org/plosbiology/s/animal-research#loc-non-human-primates

Non-human primate studies must be performed in accordance with the recommendations of the Weatherall report "The use of non-human primates in research". Manuscripts describing research involving non-human primates must include details of animal welfare, including information about housing, feeding, and environmental enrichment, and steps taken to minimize suffering, including use of anesthesia and method of sacrifice if appropriate. 

DATA POLICY:

Regardless of the method selected, please ensure that you provide the individual numerical values that underlie the summary data displayed in all relevant figure panels as they are essential for readers to assess your analysis and to reproduce it. We note that this has been done for many of the figures but request this also be done for:

*SuppFig1,7,8

DATA NOT SHOWN?

Reviewer remarks:

---

## [Editor Report · Decision Letter 3]

22 Apr 2022

Dear Xioa-Ping and Xiaoqin,

On behalf of my colleagues and the Academic Editor, Manuel Malmierca, I am pleased to say that we can in principle accept your Research Article "Distinct neuronal types contribute to hybrid temporal encoding strategies in primate auditory cortex" for publication in PLOS Biology, provided you address any remaining formatting and reporting issues on the production end. These will be detailed in an email that will follow this letter and that you will usually receive within 2-3 business days, during which time no action is required from you. Please note that we will not be able to formally accept your manuscript and schedule it for publication until you have completed any of their requested changes.

PRESS

Thank you again for choosing PLOS Biology for publication and supporting Open Access publishing. I look forward to us publishing your study, and to future interactions on other work from your laboratory. 

Sincerely, 

Kris

Kris Dickson 

Senior Editor 

PLOS Biology

kdickson@plos.org